# Dispersed emergence and protracted domestication of polyploid wheat uncovered by mosaic ancestral haploblock inference

Zihao Wang [1], Wenxi Wang [1], Xiaoming Xie[1], Yongfa Wang[1], Zhengzhao Yang[1], Huiru Peng [1,2], Mingming Xin[1,2], Yingyin Yao[1,2], Zhaorong Hu [1,2], Jie Liu [1,2], Zhenqi Su[1,2], Chaojie Xie[1,2], Baoyun Li[1,2], Zhongfu Ni [1,2], Qixin Sun [1,2✉] & Weilong Guo [1,2✉]

Major crops are all survivors of domestication bottlenecks. Studies have focused on the genetic loci related to the domestication syndrome, while the contribution of ancient haplotypes remains largely unknown. Here, an ancestral genomic haploblock dissection method is developed and applied to a resequencing dataset of 386 tetraploid/hexaploid wheat accessions, generating a pan-ancestry haploblock map. Together with cytoplastic evidences, we reveal that domesticated polyploid wheat emerged from the admixture of six founder wild emmer lineages, which contributed the foundation of ancestral mosaics. The key domestication-related loci, originated over a wide geographical range, were gradually pyramided through a protracted process. Diverse stable-inheritance ancestral haplotype groups of the chromosome central zone are identified, revealing the expanding routes of wheat and the trends of modern wheat breeding. Finally, an evolution model of polyploid wheat is proposed, highlighting the key role of wild-to-crop and interploidy introgression, that increased genomic diversity following bottlenecks introduced by domestication and polyploidization.

[1] State Key Laboratory for Agrobiotechnology, Key Laboratory of Crop Heterosis Utilization (MOE), Beijing Key Laboratory of Crop Genetic Improvement, China Agricultural University, Beijing 100193, China. [2] Frontiers Science Center for Molecular Design Breeding, China Agricultural University, Beijing 100193, China. ✉email: qxsun@cau.edu.cn; guoweilong@cau.edu.cn

Genomic introgression, as a result of natural hybridization, is a creative evolutionary force in plants[1]. As one of the staple crops worldwide, the reticulate evolutionary nature of wheat has long been recognized, and pervasive introgressions have been reported among (sub-)species[2]. Recent studies showed that bread wheat received composite introgression from tetraploid relatives[3–5], facilitating its adaptation to diverse environments and resistance to biotic stress[5], by bringing allelic resources available[6]. The genetic mechanism of the reticulate evolution is crucial to understand the domestication process of polyploid wheat and the origin of their broad adaptation.

The polyploid wheats have a complex evolutionary history[7]. The domestication of tetraploid wild emmer wheat (*Triticum turgidum ssp. dicoccoides*, BBAA) is supposed to have occurred at the onset of agriculture ~10,000 years ago in the Fertile Crescent[8,9]. Shortly after this time, hexaploid wheat (*Triticum aestivum*) was considered to originate from the hybridization between domesticated tetraploid wheat (*Triticum turgidum*, BBAA) and wild goatgrass (*Aegilops tauschii*, DD)[10–12]; it then took over tetraploid wheat and spread worldwide, arriving in Europe ~8,000 years ago[13] and in China ~4,600 years ago[14], successfully becoming a worldwide staple crop. During this domestication process of polyploid wheats, interploidy hybridization is supposed to be involved in the origins of at least two subspecies of *T. turgidum* and two subspecies of *T. aestivum*[2]. Diversity level of ancestries was high even within subspecies, as showed by the putative multiregional origin hypothesis of the domesticated emmer wheat[9,15]. Population-scale whole genome sequencing study has showed that the tetraploid donors contributed more than 20% of the genome in most hexaploid cultivars[3]. Introgressions from more distant species might introduce structural variation[4,16], some of which were valuable for agronomic practices[17]. However, the current model of wheat evolution has not fully explained the reticulate evolutionary process among *Triticum* species[18,19], especially the subspecies-level relationship between tetraploid and hexaploid wheat, and the role of admixed ancestry in the domestication, origin and diversification of polyploid wheat remains largely unknown.

Ancestries admixture detection is difficult for species of complex evolutionary history, for which three main categories of methods have been developed. The divergence-based methods, such as $RND_{min}$[20] and $G_{min}$[21], that can be calculated for individual locus, while they were unsuitable for complex evolutionary scenarios. The phylogenetic relationship based methods, like $f_d$[22], benefited from the prior biological knowledge, but their premise that the populations used as outgroup and reference have uniform ancestry might be over-simplified. The ancestry deconvolution methods, like HAPMIX[23], could deal with multiple-way admixture by evaluating ancestry across admixed genomes based on statistical models. But the data types they require for input are more specific, such as phased haplotypes and/or reference panels. Furthermore, coalescent simulations of demographic models could be used to estimate introgression rate as a component of broader demographic history, though they tended to report inaccurate estimation of ancestral state when that the demographic history of species is relatively complex[24]. The recently published genome assemblies[25] offered the opportunity to address questions through genome alignment strategy[26], though the lack of sufficient data hindered their application at the population scale to date. To fill the gaps between the complex genomic architectures and the unresolved domestication process of polyploid wheat, a method that could dissect ancestry genomic blocks utilizing the massive genome-wide sequencing data is needed.

Here, we report a systematic genomic ancestry dissection study based on a resequencing panel of interploidy wheat accessions with an algorithm designed for accurately dissecting ancestral introgression blocks. By exploring the inferred mosaic pan-ancestry haploblock map, we study the distribution of ancestry introgression blocks in different taxa of polyploid wheats. Together with cytoplasmic evidence, we discuss the implications of wild-to-crop introgression and interploidy introgression on the origin, domestication, and diversification of polyploid wheat.

## Results

**Genomic segments of high variant density attributed to wild-to-crop introgression.** To fully capture the ancestral introgression events in the wheat genome, we collected a panel of wheat accessions around the world, with 158 tetraploid and 228 hexaploid wheat accessions, covering wild emmer (WE), domesticated tetraploid (DT), hexaploid landrace (LR) and hexaploid cultivar (CV) (Supplementary Table 1, Supplementary Data 1)[3,4,25,27–29]. Moreover, five *Aegilops tauschii* accessions were included to investigate the D subgenome[4]. In total, ~271 million single nucleotide polymorphisms (SNPs) and small insertions/deletions (InDels) were identified based on these whole-genome resequencing data.

Among the A, B and D subgenomes of wheat, introgressions have been reported to be pervasive in A&B subgenomes, and it is positively correlated with genetic diversity[3]. We quantified genetic distances for several accession pairs using a sliding window approach. The results showed a mosaic of genomic blocks with high-density variants distributed along chromosomes in the A&B subgenomes (Fig. 1a), consistent with patterns derived from previous genome alignment-based study[26]. However, few high-density blocks were observed on chromosomes in the D subgenome (Supplementary Fig. 1). Interploidy genomic introgression has been reported to increase the genetic diversity in the A&B subgenomes of wheat[3]. Thus, we hypothesized that the mosaic of high-density blocks presented the antiquity of the A and B genomes as represented in the diversity in wild populations, which have become incorporated into the domesticated populations through the early admixture among lineages and the following interploidy introgression. We profiled the distributions of logged pairwise genetic distances by bin and observed bimodal distributions in A&B subgenomes of hexaploid wheat and domesticated tetraploid wheat (Fig. 1b, Supplementary Fig. 2, Supplementary Table 2). The lower-density subdistribution was shared by all three subgenomes in domesticated wheat, presenting relatively recent divergence, with an estimated range of 3,000~13,000 years before the present (yr B.P.) (Fig. 1b), which covers the putative time of wheat domestication[7]. The corresponding peak time in hexaploid wheats (~6,300 yr B.P.) was slightly later than in DTs (~7,100 yr B.P.), lending support to the emergence of hexaploid wheat after the domestication of DT. The higher-density subdistribution unique to the A&B subgenomes had an estimated time of ~70,000 yr B.P. (Fig. 1b), likely corresponding to introgressions from diverse lineages of WE[8] (Fig. 1c). A single higher-density distribution was observed for the WE and Aegilops groups (Supplementary Fig. 2), indicating rare introgression events among wild lineages. The unimodal distribution in the D subgenome of hexaploid wheat (Fig. 1b, Supplementary Fig. 2) indicates rare diversity of D subgenome involved in the origin by hexaploidization (Fig. 1c), although low-frequency subsequent introgression from other Aegilops lineages was reported[30,31]. Above all, the high diversity of the A&B subgenomes in domesticated wheat genomes is mainly attributed to the mosaic of introgression blocks, which could be traced back to specific lineages of WE (Fig. 1c).

**Dissecting the mosaic ancestry introgression blocks via Intro-Blocker.** The majority of windows present bimodal distributions for pairwise genetic distances (Supplementary Fig. 3), indicating

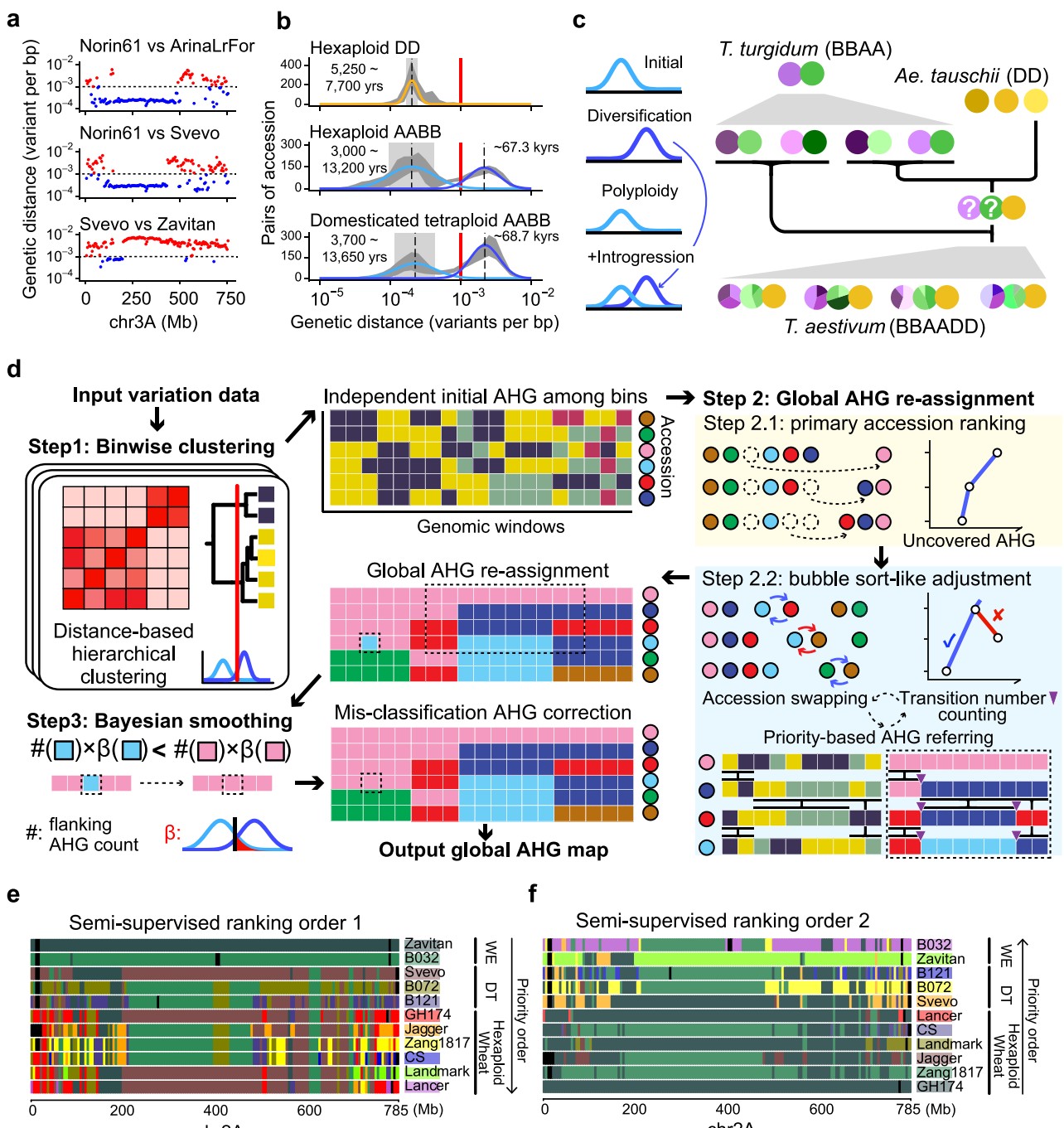

**Fig. 1 Rationale and schematic overview of the ancestry block inference algorithm IntroBlocker. a** The binwise densities of pairwise genetic differences along chromosome 3A for the three accession pairs: (1) hexaploid wheat cultivars Norin61 versus ArinaLrFor, (2) Norin61 versus tetraploid durum wheat Svevo, (3) Svevo versus tetraploid wild emmer wheat Zavitan. Each point represents a 5 Mbp genomic window. **b** Distribution of pairwise genetic distances for the D subgenome of hexaploid wheat (upper), and the A&B subgenomes in hexaploid wheat (middle) and in domesticated tetraploid wheat (bottom). Shadow ribbons, mean±sd. Solid lines represent the Gaussian distributions fitted by the EM algorithm from 1000 randomly chosen sample pairs in each track. Mean±sd of the fitted Gaussian distribution are indicated by dashed lines and shadow boxes, with corresponding divergence times labeled. Red line, the threshold at $10^{-3}$ variant per bp. **c** A model for stratified variant density in the A&B subgenomes. The interploidy introgression from highly diversified tetraploid wheat (BBAA) introduced high-diversity blocks to hybridized hexaploid wheat (BBAADD), leading to bimodal distribution in A&B subgenomes and unimodal distribution in the D subgenome. **d** Schematic overview of the IntroBlocker algorithm. Step 1, for each 5 Mbp window, the samples hierarchically clustered by genetic distance were grouped with a cut-off at the threshold ($10^{-3}$ variants per bp) imputed from a bimodal distribution. Step 2, AHGs were re-assigned globally using an accession priority-based referring strategy to minimize the transition number, where the priority order was determined by a bubble sort-like method. Step 3, a Bayesian approach was introduced for smoothing noisy signals by correcting misassigned blocks based on AHG types of neighboring windows. **e**, **f** Two colored mosaic graphs of AHGs in chromosome 2 A across 11 selected samples under a semi-supervised mode with different priority orders, WE > DT > HW (**e**) and HW > DT > WE (**f**). For each window, the same color indicates the same AHG and black indicates CNV block. The sample-dominated colors are marked under accession-IDs. WE, wild emmer wheat. DT, domesticated tetraploid wheat. HW, hexaploid wheat. Source data are provided as a Source Data file.

pervasive introgression across the chromosomes in the A&B subgenomes. To dissect the genomic ancestry of introgression blocks, we propose a semi-supervised algorithm, IntroBlocker, that incorporates the threshold for genetic distances estimated from the bimodal distribution and assigns accessions into ancestral haplotype groups (AHGs) in a binwise manner (Fig. 1d). A bin size of 5 Mbp was used in this study according to the high LD decay distances estimated in chromosomal compartments (Supplementary Fig. 4). Step 1, hierarchical clusters are built based on pairwise genetic distances for each window independently, and initial AHGs are assigned by the hard threshold imputed from the bimodal distribution. Step 2, based on the premise that the ancestry of adjacent genomic regions tends to be the same, AHGs are re-assigned globally to minimize the transition number between adjacent bins under the accession-priority-based AHG referring strategy. For each accession along with the priority order, if its windows are of the same initial AHG type with a high-priority accession, the AHG type of high-priority accession is assigned to the current accession. Otherwise, a novel AHG type is assigned and dominated by the current accession (see Methods for details). The priority order of accessions is determined by a bubble sort-like method consisting of two parts. Step 2.1 aims to select a subset of accessions covering the majority of AHG diversity (95% in default). Based on this primary order, step 2.2 aims to find the order that could minimize the transition number by swapping accessions iteratively. In each iteration, the order of adjacent accessions is swapped, AHGs are re-assigned using the current order and the transition number is counted. The swapped order is reserved if the transition number decreases, and swapping continues until no changes could be made. This ranking algorithm could be evoked globally (un-supervised mode), and it also could be evoked within each accession group assigned based on prior knowledge (semi-supervised mode, Supplementary Fig. 5). Step 3, a Bayesian method was evoked to smooth the potentially misassigned blocks under the hard threshold by referring to the AHG types in neighboring windows (Supplementary Fig. 6).

We validated the reliability of IntroBlocker in both simulated and real dataset. In various simulated admixture scenarios, IntroBlocker achieved the accuracy of 97.0% on average (Supplementary Fig. 7, Supplementary Fig. 8). We further compared IntroBlocker with the haplotypes detected by the $f_d$ statistic[3,22]. The results obtained through two methods are basically conformable (Supplementary Fig. 9). Furthermore, IntroBlocker could leverage prior knowledge of both biological and evolutionary in a flexible manner. Using a panel of ten assembled wheat accessions as an example, threaded AHGs were inferred under un-supervised mode, and the results indicated pervasive heterogeneous ancestries as a result of introgression and large AHG blocks around the pericentromeric region (Supplementary Fig. 10). For the same set of underlying AHGs, the threading could be different according to the priority order. A panel of interploidy accessions was demonstrated with the order of "WE > DT > hexaploid wheat" on chromosome 2A, and the blocks with mosaic ancestry patterns were derived from two wild emmer accessions and then transferred to the durum wheat Svevo (Fig. 1e). With a reversed order of "hexaploid wheat > DT > WE", the mosaic genomic blocks shared by hexaploid accessions were traced to DT and WE, revealing the recombination pattern that formed modern hexaploid wheat accessions (Fig. 1f). The consistency of the inferred AHGs and the flexibility of their presentations allows IntroBlocker to disentangle ancestral introgression blocks throughout tetraploid and hexaploid wheat.

**AHG reveals gene flow from wild emmer to hexaploid wheat.** To systematically characterize ancestral introgression in wheat,

we constructed a pan-ancestry haploblock map by applying IntroBlocker to 386 accessions and inferred AHGs for all 5 Mbp nonoverlapping windows (Supplementary Data 2, Supplementary Fig. 11). The population structure derived from AHG-based distances resembled the SNP-based phylogenetic relationship for tetraploid and hexaploid wheat[3], suggesting that the differences in ancestral blocks were major source of genetic differentiation among samples (Fig. 2a, Supplementary Fig. 12). The AHGs in the pan-ancestry map were investigated in the WE, DT, LR, and CV groups, to explore the gene flow and gene pool transitions among polyploid wheat. Chromosome-wise analysis showed that AHGs are less diverse in the centromeric region (Fig. 2b), coincident with a lower recombination rate in centromeric regions of wheat chromosomes[32]. Rare introgression blocks were found in the D subgenome, except for a few hotspots, with the dominant AHG accounting for 95.4% of the D subgenome (Supplementary Fig. 13, Supplementary Data 3).

Overall, the WE group preserved the most diverse AHGs, with an average of ~20 per window (Fig. 2c), although the saturation curve indicated that this number was underestimated due to the limited number of WE accessions collected (Fig. 2d). The other groups showed significant reductions in diversity, with only ~5 AHGs remaining in hexaploid wheat (Fig. 2c). The cumulative frequency distribution showed that 2-6 common AHGs (detection frequency ≥5%) contributed nearly half (48.1%) of each chromosome in the A&B subgenomes (Fig. 2e). We showed that four groups shared 50.6% of all AHGs, and an additional 21.7% were shared by three consecutive groups (Fig. 2f, Supplementary Table 3), indicating the continuous transition of gene pools throughout domestication and hexaploidization. A total of 19.6% of AHGs were shared by DT, LR, CV, in contrast with 1.1% shared by WE, LR, and CV, supporting DT as the gene pool that directly contributed AHG introgression into the gene pool in hexaploid wheat. Overall, the AHG-based evidence implied the gene pool of hexaploid wheat could be traced back to that of DT and eventually to that of WE, implying continuous gene flow from tetraploid wheat to hexaploid wheat.

**Tracing the founder wild emmer lineages that contributed to domesticated polyploid wheat.** Wild emmer wheat is regarded as the progenitor of domesticated tetraploid wheat[7], and accumulated evidence indicates that pervasive introgression in hexaploid wheat occurs from wild emmers[3–5], implying a complicated phylogeny. We computed the AHG-based contribution of each WE accession to the domesticated polyploid wheat gene pool and identified a total of 12 WE accessions in the Fertile Crescent, belonging to six lineages (Supplementary Data 1), that were basal to DT (Fig. 3a). The accessions in these six lineages contributed significantly more AHG blocks (two-tailed $t$-test, $P = 4.8 \times 10^{-11}$) to domesticated polyploid wheat (23.3%) than other WE accessions (4.6%), and a significantly higher proportion of genomewide AHGs (two-tailed $t$-test, $P = 1.9 \times 10^{-11}$) from the six lineages could be detected in domesticated polyploid wheat (62.7%), than other WE accessions (17.0%), suggesting their role as founder donors in the early formation of DT (Fig. 3a). These founder lineages collectively explained 53.5% of the AHGs in domesticated polyploid wheat, while the contributions varied among chromosomes (Fig. 3b). Generally, the DT accessions presented a highly mosaic pattern of AHGs along chromosomes, in contrast with rare AHG introgression between WE lineages (Fig. 3c), suggesting extensive wild-to-DT introgression in the initial domestication of emmer wheat under early cultivation in the Fertile Crescent. Meanwhile, our results showed that WE accessions have almost equal contributions to the gene pool of DT and hexaploid wheat (Supplementary Fig. 14), suggesting that

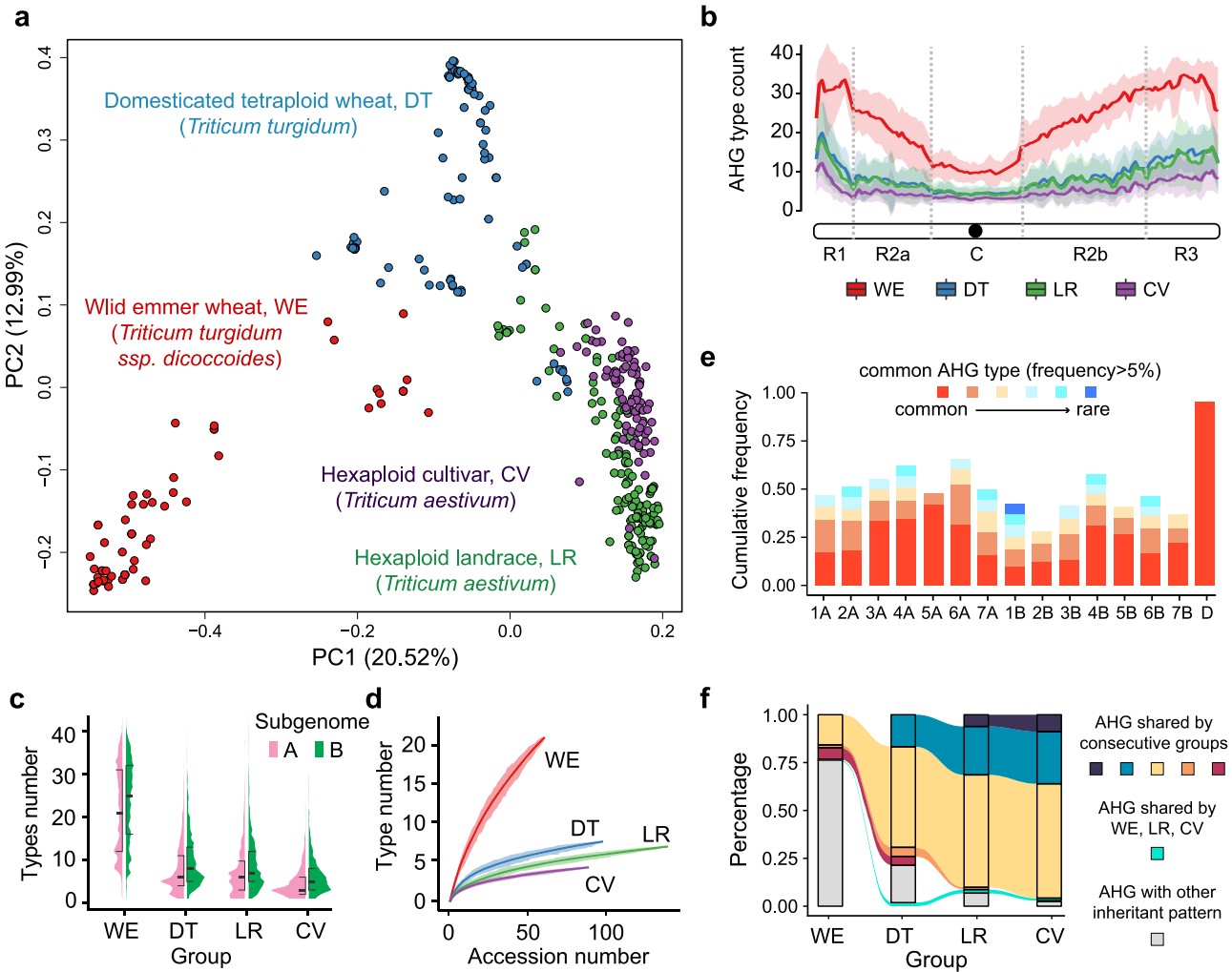

**Fig. 2 AHG reveals the transition of gene pools in polyploid wheat. a** Principal component analysis plot constructed with AHG-based distances for all 5-Mb windows in A&B genomes. **b** Counts of AHG types across chromosome model of A&B subgenomes. Solid line and colored area, mean±sd. WE, wild emmer wheat. DT, domesticated tetraploid wheat. LR, hexaploid wheat landrace. CV, hexaploid wheat cultivar. The chromosomal zones R1, R2a, C, R2b and R3 are consistent with previous publication[32]. **c** Violin plot showing the count distributions of AHG types in the A&B subgenomes for each group. **d** Saturation curves showing the cumulative number of AHG types versus the number of accessions included. Solid line and shaded region, average and 90% confidence intervals for 14 A&B chromosomes. **e** Stacked bar graph showing the frequency and corresponding order of common AHGs (frequency > 5%) by chromosome. "D" indicates the average value of 7 chromosomes in D subgenome. **f** Proportion of shared AHGs among four taxonomic groups shown by alluvial plot revealed genetic flows through the route of WE-DT-LR-CV. A dramatically higher proportion of AHGs in the hexaploid groups can be traced back to the DT group than to the WE group using the collected samples, indicating DT as the closest gene pool for hexaploid wheat. Source data are provided as a Source Data file.

no substantial wild-to-crop introgression occurred in the hexaploid wheat. Overall, our results show that domesticated polyploid wheat originated from limited lineages of WE, and the wild-to-crop introgression contributed to the initial landscape of mosaic AHG patterns in the A&B subgenomes of wheat.

**Tracing the origin of hexaploid wheat.** Hexaploid wheat is thought to emerge from the hybridization between domesticated tetraploid wheat *T. turgidum* (AABB) and the diploid species *Aegilops tauschii* (DD)[33], while solid genetic evidence implying which tetraploid subspecies involved is lacking. To further trace the origin of hexaploid wheat, we examined the AHGs in hexaploid landraces that were shared with various DT subspecies, and the results showed that rivet wheat (*T. turgidum L. ssp. turgidum*) shared high proportion of its gene pools with LR (Supplementary Fig. 15), consistent with their broader tolerance range[34]. To obtain evidence of cytoplasmic variations, we

identified 165 high-confidence variants in chloroplast genome sequences from 306 wheat accessions (Supplementary Data 4), most of which persisted only in WE accessions, while only 27 variants existed in hexaploid wheat. In the cytoplasmic variation-based phylogenetic tree (Fig. 3d), the WE accessions were the most diverse, with most DT accessions forming a single branch, except for 4 DT accessions that clustered in other branches, reflecting potential wild to domesticated emmer introgression as previously reported[9]. The LR hexaploid accessions were nested with several DT lineages, confirming that hexaploid wheat emerged from the DT population and through further hybridization events with either tetraploid or hexaploid wheat as the reciprocal maternal line. Interestingly, the widely planted durum wheat (*T. turgidum ssp. durum*)[35] was distributed in most branches of DT, indicating pervasive hybridization with other subspecies of DT. Notably, we found durum wheat in all five main clades of the hexaploid subgroup (Fig. 3d), implying that the

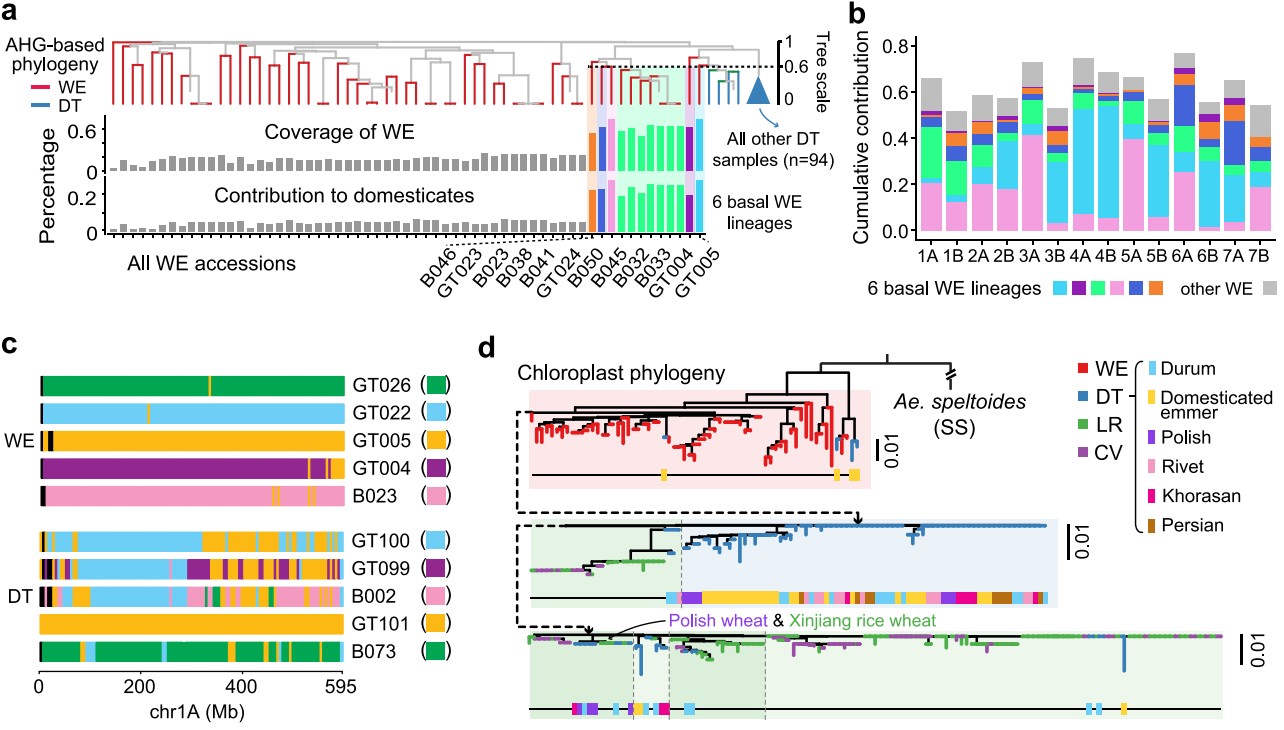

**Fig. 3 Tracing the basal lineages for wheat domestication. a** NJ tree of WE and DT accessions constructed with AHG-based distances for the A&B subgenomes. The six basal lineages are noted by colored areas, and branches of DT accessions are collapsed. The percentage of AHGs in domesticated wheat gene pools that were found in each WE accession (contribution) and the genomic proportion of each WE accession that contributed AHGs to domesticated wheat gene pools (coverage) is shown. 6 basal lineages, consisted of 12 accessions, are highlighted. **b** Cumulative contribution of 6 WE founder lineages and other WE accessions to the domesticated wheat genetic pools is shown for individual chromosomes. **c** Mosaic graphs of AHGs for 5 WE and 5 DT accessions across chromosome 1 A. **d** Chloroplast SNP based NJ-tree provides cytoplasmic evidence for the origin of wheat. The tree is rooted by assigning *Ae. speltoides* (SS) accessions as an outgroup. Main clades are separated by gray dash lines. Clades containing *T. polonicum* (Polish wheat, tetraploid) and *T. petropavlovskyi* (Xinjiang wheat, hexaploid) were labeled, consistent with reported genomic introgression among the two populations[28]. Source data are provided as a Source Data file.

durum wheat was likely to play a more important role in the origin and further spreading process of hexaploid wheat among the subspecies of domesticated tetraploid wheat. Additionally, *T. polonicum* (Polish wheat, BBAA) and *T. petropavlovskyi* (Xinjiang rice wheat, BBAADD) accessions showed close cytoplasmic relationships, indicating potential maternal inheritance between the two subgroups, consistent with known interploidy introgression events between them[28]. The combined evidence from nucleus and cytoplasm indicates that the origin of hexaploid wheat was nested within DT, followed by pervasive interploidy introgression.

**The key domestication loci were pyramided from heterogeneous donors and fixed through a protracted process.** To investigate the dynamic changes in binwise AHGs in the process of wheat domestication, we proxied the Shannon diversity index (H) of AHGs for all 5 Mbp windows across the A&B subgenomes between taxonomic groups at three stages (WE-DT, DT-LR, LR-CV; Supplementary Fig. 16). A total of 176 windows were detected under continuous selection in at least two stages (Supplementary Fig. 17, Supplementary Data 5), and the majority of homoeologous loci harbored signals of asynchronous selection (Supplementary Fig. 18). A total of 256 windows (50, 123, and 83 in three stages) were shown to be fixed for selected AHGs (H < 0.05), which are considered to be potential domestication-related loci (Supplementary Data 6); these loci containing some known genes involved in the domestication syndrome, such as *TaPpd-2A*[36], *TaGI-3A*[37] (photoperiod), and *TaGS5-3A*[38] (grain size) (Fig. 4a).

The domestication loci considered to be under different selection pressures were fixed stepwise in the domestication and breeding process of wheat. The genes controlling rachis brittleness, *TaBtr1-3A/-3B*[3,39], were both fixed at the earliest transition from WE to DT. In a subsequent diversification of durum wheat from domesticated emmer, traits for soft glumes and non-hulled seeds emerged[7], and the loci surrounding the *TaTg-2B*[40] and *TaQ-5A*[41,42] genes were almost fixed (Fig. 4b, Supplementary Fig. 19–22). Despite finding a total of 24 AHG types in the window surrounding *TaBtr1-3B*, only one type became almost fixed in DT; this type could be traced back to 2 WE accessions, GT004 and GT005, that were supposed to be sequence template donors for the domesticated form of *TaBtr1-3B*. In addition, the hitchhiking effect fixed a significantly larger flanking region in DT than in LR and CV (Supplementary Fig. 23). The evolutionary diagram of the 1 Mbp region around *TaBtr1-3A* mirrors that around *TaBtr1-3B*, except for the higher ratio of the prevailing type in WE. Although not officially cloned[40,43], we could infer that the region surrounding *TaTg-2B* was under continuous selection, as the percent of modern AHG types increased from 1.6% in WE to 35.3% in domesticated emmer and became fixed in all durum wheat accessions. Similarly, 1 of the 42 AHGs in the *TaQ-5A* region was continuously selected, with a frequency reaching 91.0% in durum wheat and finally became fixed in all cultivars of common wheat. However, its donor WE accession, GT028, was not from the founder lineage and was located in Bahrain, which is outside the core Fertile Crescent area. Interestingly, only two minor AHG types in the *TaQ-5A* region in landraces existed in spelt wheat and Yunnan hulled wheat, both showed a hulled spike morphology[44].

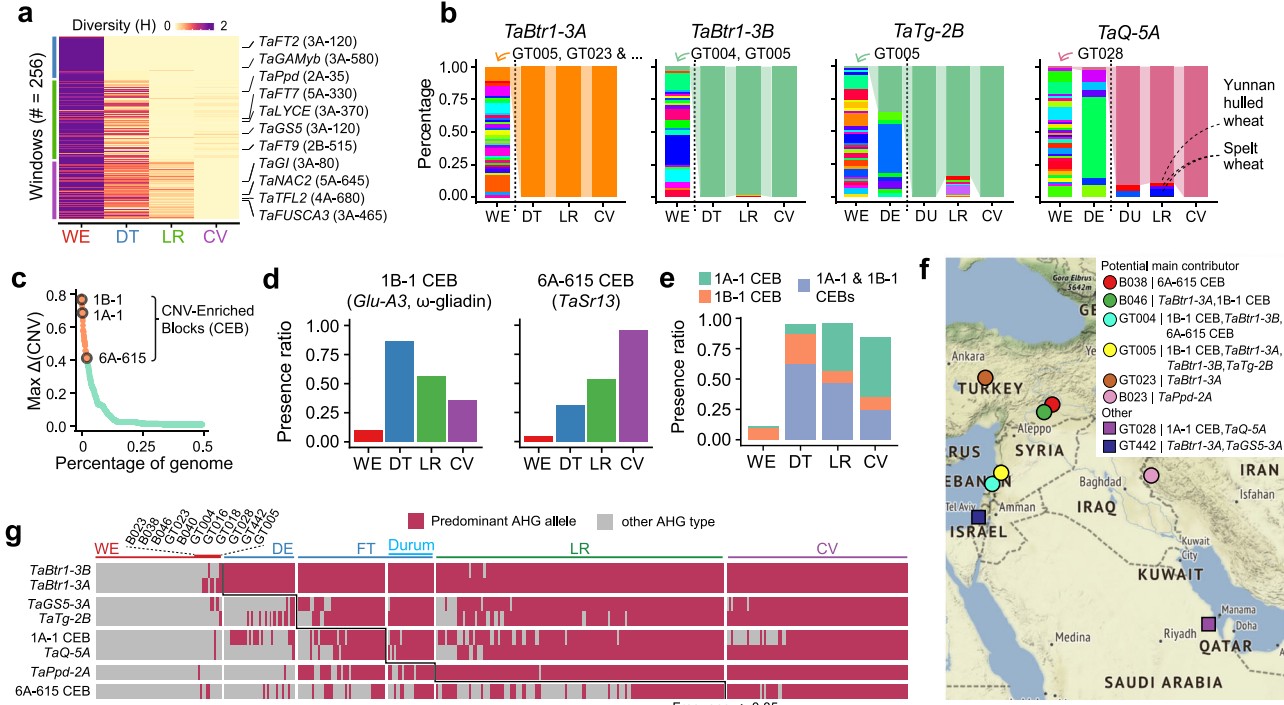

**Fig. 4 Heterogeneous origins and stepwise accumulation of key domestication-related loci. a** Heatmap illustrating the stepwise fixation of AHGs across 256 windows. Windows were sorted to match their relative order of fixation, and three major groups emerged, consistent with the groups in which fixation was first reached. Representative windows were labeled with residing known adaptive genes in the same manner as **a**. **b** Dynamics of AHG frequencies for key domestication-related loci. The selected AHGs were connected through taxonomic groups with colored ribbons. WE accessions carrying selected AHG types were labeled with IDs. Due to space limitation, the full accession list for *TaBtr1-3A* is shown in Supplementary Data 8. The window size is 5 Mbp by default. The window size for *TaBtr1-3A* was set to 1 Mbp to match its relatively limited selective sweeps. **c** Distribution of MaxΔ(CNV) for 5 Mbp windows (points) with CNVs detected in at least one accession. MaxΔ(CNV): max CNV block ratio difference between taxonomic groups. Orange, CNV enriched block (CEB) with Max Δ(CNV) > 0.4. Selected top-ranking CEBs were labeled in the same manner as **a**. **d** Frequency of the presence of 1B-1 CEB (left) and 6A-615 CEB (right) in four taxonomic groups. **e** Frequency of the presence of 1A-1 CEB alone (green), 1B-1 CEB alone (orange) or concurrence (blue) in four taxonomic groups. **f** Geographical location of WE accessions contributing predominant AHG alleles of domesticated loci and selected CEBs. **g** Presence and absence diagram of the predominant AHG type in domestication and adaptation genes showing the pyramiding process throughout evolution. Map in **f** was generated with geographic information from the Stamen Map. Source data are provided as a Source Data file.

We also investigated copy number variation enriched blocks (CEBs) with respect to the CS reference genome; these blocks comprised a substantial proportion of the wheat genome[16] and may be ignored in AHG-based studies. A total of 46 windows were classified as selected CEBs, and the changes in their frequencies across groups were ≥ 0.4 for at least one stage (Fig. 4c, Supplementary Data 7). Two top-ranking colinear regions, the first 5 Mbp windows in both chromosome 1 A and 1B (noted as 1A-1 CEB and 1B-1 CEB), which are known to contain baking quality trait-related genes, such as *Glu-A3* and ω-gliadin gene cluster[45,46], show a low frequency in WE, while their frequency dramatically increased in domesticated groups (Fig. 4d, Supplementary Fig. 24). Only 11.5% of WE accessions were found to have at least one of the two colinear CEBs, in contrast with 84.1% to 95.7% of accessions in the three domesticated groups (Fig. 4e), indicating that the presence of these CEBs may play an essential role in the end-use quality of wheat. Another top-ranking CEB, 6A-615, containing the stem rust resistance gene *TaSr13*[47], showed a continuously increasing trend during the evolution process (Fig. 4d). These selected AHGs revealed the presence-absence dynamics of genomic blocks, adding evidence for domestication-related loci in wheat.

Notably, the selected domestication-related loci can be traced to allopatric WE accessions in various lineages with a dispersed distribution in the core area of the Fertile Crescent, such as Turkey, Syria, and Israel (Fig. 4f), raising the question of how

hexaploid wheat assembled all of its domestication-related loci. The presence of key domestication-related loci was profiled in accessions throughout the evolutionary process, and the results indicate that these loci gradually accumulated and were fixed in the population through a protracted process (Fig. 4g, Supplementary Data 8). *TaBtr1-3A/-3B* were first fixed in domesticated emmer, and then *TaTg-2B* and *TaGS5-3A* quickly accumulated and were almost fixed in free-threshing tetraploid wheat. 1A-1 CEB and *TaQ-5A* were almost fixed in durum wheat. Hexaploid wheat inherited these preadapted blocks from its tetraploid contributor, and as such, the domesticated gene *TaPpd-2A* was fixed. The *TaSr13* linked 6A-615 CEB was almost fixed in CV. In summary, the combined geographical and phylogenetic evidence showed heterogeneous origins of domesticated-related loci and suggested that domestication was a protracted process, with the origin of hexaploid wheat nested in the domestication of tetraploid wheat.

**The stable centAHG acts as the backbone of the chromosome during evolution.** Several large introgression blocks in pericentromeric regions have been reported[3,5,26,48], and the chromosome-scale mosaic pan-ancestry haploblock map also revealed large stable AHG blocks in the central zone of the chromosome (centAHG) (Supplementary Fig. 10). To examine the patterns of centAHGs, we proposed a strategy to determine the centromeric

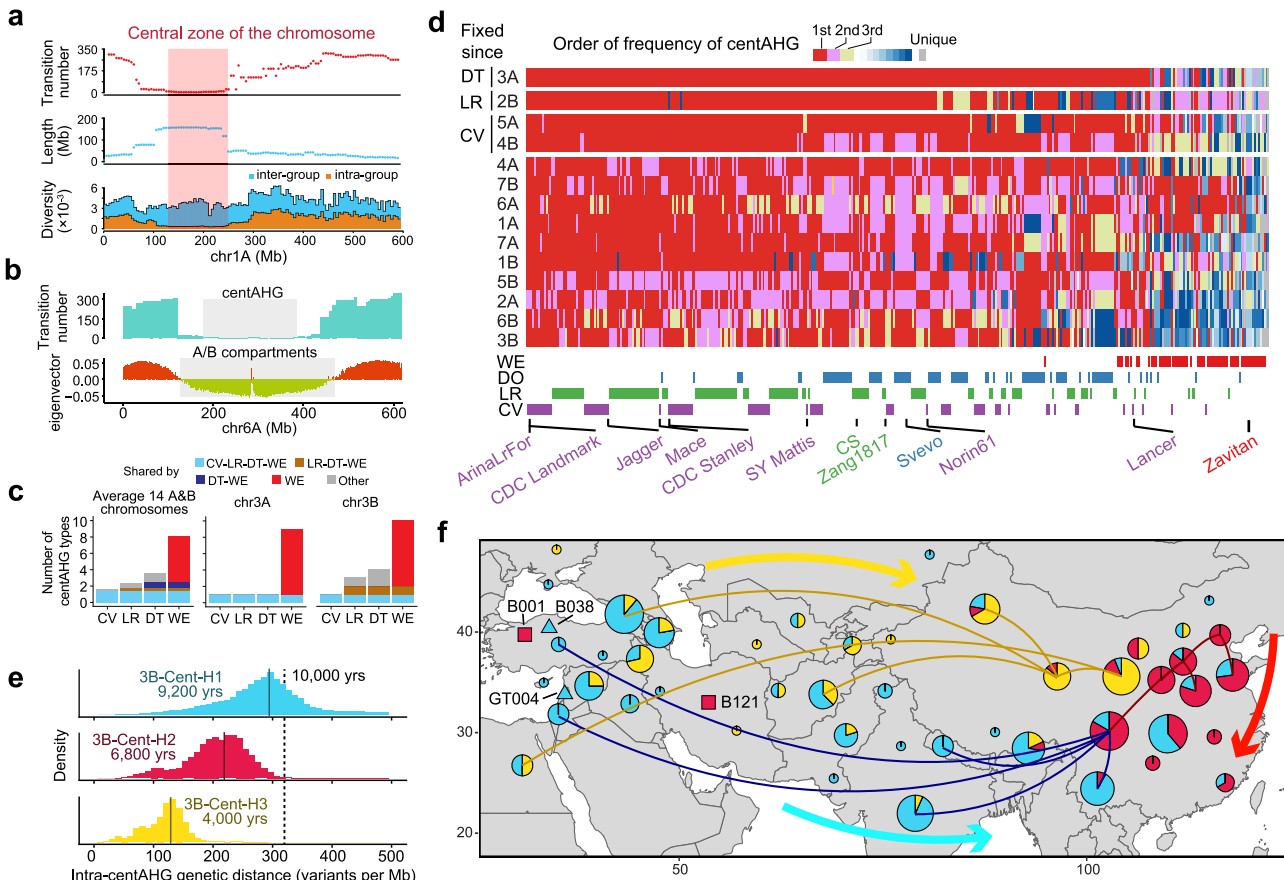

**Fig. 5 Centromeric ancestral haplotype group (centAHG) acts as the chromosome backbone in evolution. a** Positioning of centAHGs on chromosome 1 A, based on the time of transition among AHG types between adjacent windows (upper), length of consecutive windows of the same type of AHG (middle) and ratio between inter- and intra- AHG type genetic diversity (lower) of all 386 accessions. **b** The consistency between the partition of the centAHG and the A/B compartment from a previously published study[49] on chr6A. **c** Proportion of centAHGs among four taxonomic groups that could be traced back through the route of WE-DT-LR-CV of 14 A&B chromosomes shown by stacked bar chart. **d** Distribution of centAHGs in A&B chromosomes among 386 samples. The color index corresponds to the order based on frequency of centAHGs on each chromosome. Accessions are ordered by the overall frequency of centAHGs they contain. The diversity-based order of chromosomes revealed the stepwise fixation of centAHGs. The taxonomic group of each accession and name of selected accessions are shown. **e** Distribution of pairwise genetic distances among accessions from 3 main 3B centAHGs with correspondingly estimated divergence times labeled. **f** The potential dispersal routes of the initial introduction of hexaploid wheat to East Asia. Routes (curves) were inferred from the phylogenetic relationship of the three main 3B centAHGs in landraces (pie chart) and directions (arrows) were determined by manual consolidation of the literatures. An additional 281 landraces from previously published studies with geographical information available were added[5,18,48]. The geographical locations of WE accessions carrying 3B-Cent-H1 and DT accessions carrying 3B-Cent-H2 and 3B-Cent-H3 are labeled. Map in f was generated with geographic information from the Natural Earth project database. Source data are provided as a Source Data file.

AHG blocks on chromosomes with consistent AHGs spanning from 80 to 250 Mbp (Supplementary Table 4). The results showed fewer AHG switches within the identified centAHG blocks and highly reduced intra-centAHG genetic diversity compared with inter-centAHG genetic diversity (Fig. 5a, Supplementary Fig. 25). We inferred centAHGs for all 14 chromosomes in the A&B subgenomes (Supplementary Data 9). Interestingly, the partition of centAHG was roughly consistent with the A/B compartment of the wheat genome inferred in a previous study[49] (Fig. 5b, Supplementary Fig. 26), suggesting that the formation of centAHG might be relevant to the 3D structure of genome in the nucleus. The number of centAHG types varies among chromosomes, with the highest diversity on 3B (23 types) and the lowest on 6A (10 types); the diversity of domesticated groups was largely reduced, with an average of ~3.9 on each chromosome (Supplementary Fig. 27). The centAHGs were stable during evolution, even between gene pools of tetraploid and hexaploid wheat, and 98.6% of the centAHGs in CV could be traced back to LR (Fig. 5c).

By inspecting the frequencies and distributions of centAHGs across all accessions (Fig. 5d), we found that centAHG of several chromosomes were gradually fixed during evolution, such as 3A fixed in DT, 2B fixed in LR, 5A and 4B fixed in CV (frequency > 98%). WE accessions harbored a higher frequency of rare centAHGs, suggesting that many genetic resources in wild emmer were underexplored. Modern wheat cultivars, such as ArinaLrFor and CDC Landmark, significantly accumulated the most pervasive centAHG types across all A&B subgenome chromosomes compared to DT (Wilcoxon rank-sum test, $P = 4.6 \times 10^{-9}$). Although diverse centAHGs still existed in some chromosomes, a general trend was observed in which the major centAHG of each chromosome tended to be utilized in modern wheat breeding.

Further analysis showed that the centAHG-based phylogenetic tree mirrored the genetic structures among the taxonomic groups (Supplementary Fig. 28), indicating that centAHGs could provide evidence of gene pool transitions during evolution. Two types of haplotype blocks in the centromeric regions of chromosome 3B

have been reported to lead genetic differentiation in Chinese landraces[50]. In our dataset, we found three predominant centAHGs in chromosome 3B among global common wheat landraces, while one type (3B-Cen-H2) was only found in Chinese landraces at 38.0% (Supplementary Fig. 29). To further investigate the origin of the three centAHGs, additional whole-genome resequencing[3,48] and whole-exome capture datasets[5] were included and 3B-centAHGs were characterized in an extended panel of 412 hexaploid wheat landraces (Supplementary Data 10). The geographical distributions of the three major centAHGs were profiled for possible spreading routes inferred by minimum spanning trees (Fig. 5f), and their origin times were also estimated by intra-centAHG genetic distances (Fig. 5e). Interestingly, the results indicated that the most ancient haplotype 3B-Cen-H1, which can be traced back to WE accessions gathered from Turkey and Lebanon (Supplementary Fig. 30), is the most widespread in the world, with the earliest origin time of ~9,200 years ago, matching the incipient period of hexaploid wheat (Fig. 1b). The low frequency of 3B-Cen-H1 in Chinese landraces indicated that it might be the latest centAHG introduced to China with a route that likely passed through South and Southeast Asia and then Tibet and Yunnan provinces of China. The youngest haplotype 3B-Cen-H3 (~4,000 years ago), which was mainly distributed at high latitudes, was introduced into China via Turkmenistan and Xinjiang, coinciding with the Silk Road. The Chinese landraces unique type 3B-Cen-H2 dominated landraces from the winter wheat zone in northern China. The three potential dispersal routes were roughly consistent with archeological evidence[14], suggesting that centAHGs acted as the putative backbone of chromosomes and might play an essential role in the domestication and varietal diversification of wheat.

**A genomic footprint based model for the origin and domestication process of polyploid wheat**. We propose a refined model for the origin of wheat with multilevel genetic evidence by highlighting the dispersed emergence and protracted domestication of polyploid wheat (Fig. 6). In this scenario, *T. turgidum* (BBAA) differentiated into multiple WE lineages over a long time since its origin by polyploidization, and the low frequency of gene flow allowed the accumulation of genetic diversity among the WE lineages scattered before the beginning of agriculture in the Fertile Crescent area. During the Neolithic period (~10,000 years ago), domesticated emmer emerged from the admixture of wild emmer lineages in the Fertile Crescent and introduced the 1st genomic diversity reduction, but wild-to-crop introgression by unconscious mixing between multiple WE lineages in the various regions, created the mosaic ancestral genome and remarkably increased genetic diversity. The restored diversity in domesticated emmer supports its further spread and continuous domestication, which resulted in multiple tetraploid subspecies, such as durum, rivet and polish wheat. Later, natural hybridization between free-threshing tetraploid wheat (BBAA) and wild goatgrass (DD), which was likely to occur in a durum wheat field, created the first hexaploid wheat (BBAADD). Hexaploid wheat was successfully selected and became dominant in the local field, although farmers were unaware of its ploidy. Although hexaploidization introduced the 2nd genomic diversity reduction, the interploidy introgression promised by the state of hybrid swarm with already-spread domesticated tetraploid wheat introduced ancestral genomic mosaics to hexaploid wheat. The restored genetic diversity of hexaploid wheat supports its spread and further domestication. Then, hexaploid wheat finally displaced tetraploid wheat and was grown worldwide. Modern breeding activity introduced the 3rd genomic diversity reduction, while hybridization followed by

selection counteracted the reduction by increasing the genetic combinations. In summary, we proposed that the polyploid wheat originated from a dispersed geographic range and the domestication-related loci accumulated in a protracted process, and the coexistence of domesticated tetraploid and hexaploid wheat in history entangled their domestication process and resulted in shared gene pools (Supplementary Fig. 31).

## Discussion

The question of how domesticated polyploid wheat originated is a long-standing issue[7]. Our results showed that domesticated polyploid wheat accessions presented mosaic ancestral patterns compared with wild emmer accessions, consistent with previous findings in other crops[51,52] and animals[53,54]. It has been reported that populations from both the northern and southern Levant played important roles in the formation of domesticated tetraploid wheat[15,55]. The geographical distribution of the founder wild emmer lineages, and the heterogeneous donors of important domestication-related loci and centAHG all suggested that a widely dispersed region was involved in the origin of domesticated tetraploid wheat. The reticulate process plays a key role in early domestication, as shown in rice[56] and maize[57]. The reconstructed pyramiding process of the domesticated-related loci in domesticated polyploid wheat along with human-mediated hybridization, and gradual enrichment of the gene pools during further domestication support the domestication of polyploid wheat as a continuous and protracted process, which is consistent with genomic-based and archeology-based studies[58,59] and coincident with the semidomesticated model of maize revealed by paleogenomic research[60].

In contrast to rice and maize, hexaploid wheat emerged in a field of domesticated tetraploid wheat and interploidy hybrid swarms happened in the early agricultural field[61]. This complicated picture of the early domestication process[2] implying the contributions of interploidy and interspecific gene flow to the evolution of *Triticum* species might be underestimated previously. Recent evidence of pervasive introgression in wheat[3,5] also supported that the bread wheat utilized existing gene pool of tetraploid wheat through interploidy introgression during spreading[7]. However, as it was suggested by the growing archaeogenomic and archaeobotanical evidence from various crops, populations involved in the process of domestication were likely to be large[59]. In addition, the multiple hybridization hypothesis of hexaploid wheat was proposed based on the evidence from specific genomic signatures of *Ae. tauschii* and the D subgenome of bread wheat[62,63]. Overall, it is likely that both direct inheritance of admixed ancestries through the process of domestication and interploidy introgression contributed to the origin of hexaploid wheat.

Studies in sunflower[64] and cotton[65] showed that the haplotype blocks that keep adaptive alleles together contributed to the ecotypic adaptation, the recombination rate of which were often suppressed inversions. The identified centromeric ancestral haplotype groups (centAHG) might play a similar role in wheat without large structural variants. They stretched dozens or hundreds of megabase pairs in length, containing thousands of genes. They could be stably inherited from tetraploid wheat under selection and might act as the chromosome backbones through domestication. Significant phenotypic differences have been detected between accessions with different centromeric haplotype[48], and the centromeric region of 3B chromosome has been reported to lead genetic differentiation in Chinese landraces[50]. These evidences suggested that centAHG might be instrumental in the quick adaption to local environments of wheat during spreading. Furthermore, studies in Triticeae

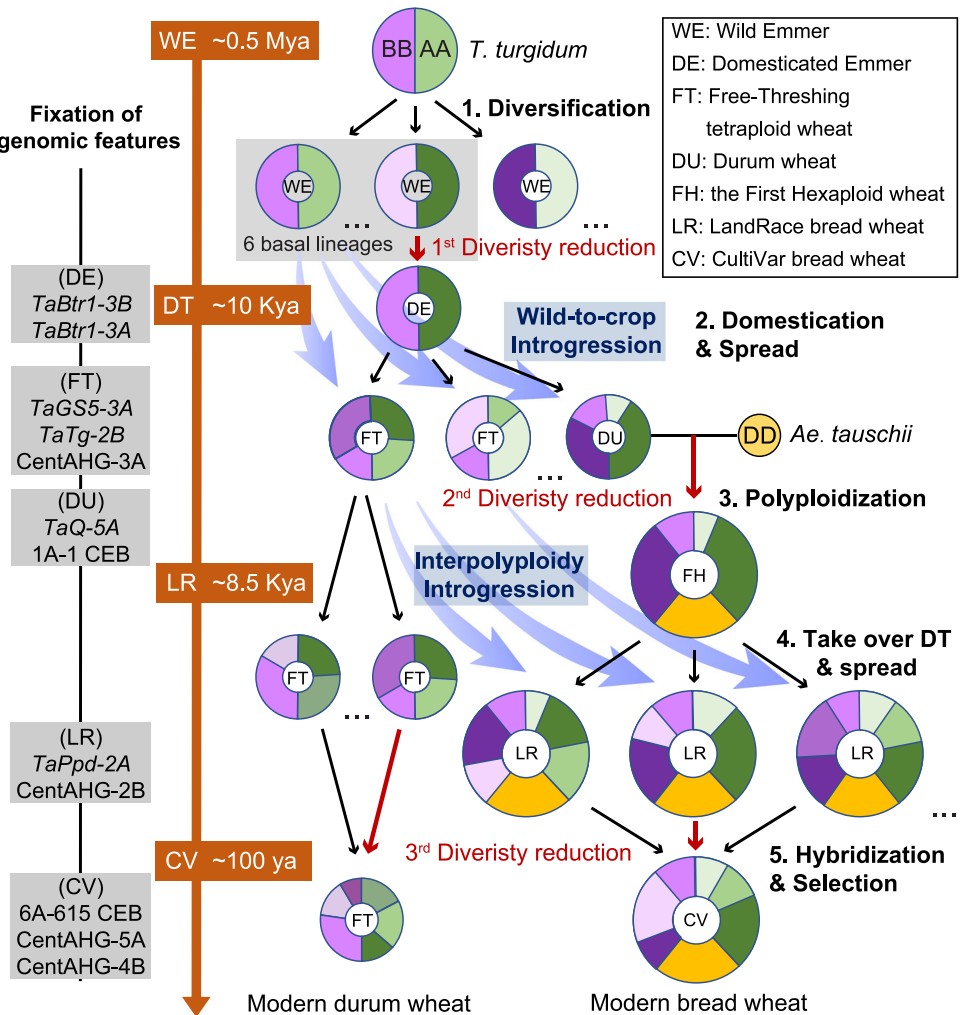

**Fig. 6 Schematic illustration of the dispersed emergence and protracted domestication of polyploid wheat.** The evolutionary scenario of modern domesticated wheat from initial tetraploid wheat (right), and the corresponding time scale (middle). Purple, green and yellow sectors denoting the A, B and D subgenomes, respectively, and the types of AHGs were discriminated by their brightness. Succinct descriptions of five major steps are denoted with serial numbers. Arrow colors indicate the phylogenetic relatedness between taxa. Black, direct pedigree. Blue, genetic introgression. Red, major genetic diversity reduction. The genomic features fixed in each group were noted (left). This schema is based on the results from this study and prior assumptions from the literature.

suggested that the homologous centromere functional sequences might be related to homolog pairing and recombination. As rare crossovers were detected inter-centAHGs, modern breeding tended to reduce the diversity of centAHGs by selecting the convergent type. It suggested that uniform centAHGs are important for increasing breeding efficiency[48].

As climate change accelerated, harnessing the adaptation mechanism of major crops to create outstanding cultivars has become necessary and urgent[66]. A greater understanding of domestication would provide a theoretical basis for how we could achieve it[67], like creating novel crops through de novo domestication, especially polyploids of vigorousness and robustness[68]. Besides the knowledge regarding wheat domestication gained during the last decade, still many questions remain unsettled. For example, the formation mechanisms and genetic relationships of subspecies in the *Triticum-Aegilops* complex could be illuminated further in the light of ancestral mosaics. Considering the current report about the reticulated evolution and frequent interploidy introgression of the bread wheat[3], the synergistic improvement of both tetraploid durum wheat and hexaploid bread wheat could be accelerated by transferring the beneficial alleles between the two genetic pools. Furthermore, a large portion of domesticated genes

within the favorable genomic segments during domestication and improvement was underexplored, and their detailed evolutionary trajectories remain unclear. Additionally, the relationship between the gene pool of *Aegilops tauschii* and the wheat D subgenome may be oversimplified[62]. With comprehensive genomic data of *Triticum-Aegilops* species available and innovations in computing algorithms, answers to these questions will be clear and ultimately used as a source of innovations for wheat improvement.

## Methods

**Sample collection and whole-genome resequencing.** We collected a panel of 393 interploidy wheat accessions comprising seven diploid accessions (five *Ae. tauschii* with DD genome and two *Ae. longissima* with SS genome), 158 tetraploid accessions (BBAA genome) and 228 worldwide hexaploid accessions (BBAADD genome). Whole-genome resequencing data of 16 accessions were generated in this study, and the other 377 accessions were collected from a total of 8 datasets[3,4,25,27–29] (Supplementary Data 1).

**Whole-genome sequencing and quality control.** Genomic DNA was extracted from young leaves of 16 accessions following a standard CTAB protocol[69]. DNA libraries were constructed by Novogene and sequenced with the Illumina Hiseq Xten PE150 platform with an insert size of approximately 500 bp at an average

depth of 5.2 (Supplementary Table 1). For each accession, the quality control for raw reads were conducted using Trimmomatic[70]. The leading and trailing low quality bases (below quality 3) were removed. Each read was scanned with a 4-base sliding window, cutting when the average quality per base drops below 15. Reads below the 36 bases long were removed.

**Genomic variant calling and quality control.** High-quality clean reads were mapped to the CS wheat reference genome (IWGSC RefSeq v1.0)[32] using BWA-MEM[71] with default parameters. Read pairs with abnormal insert sizes (>10,000 bp or < −10,000 bp or = 0 bp) or low mapping qualities (<1) were filtered using Bamtools v2.4.1[72]. Potential PCR duplicates reads were further removed using the rmdup function in Samtools v1.3.1[73]. SNPs and INDELs were identified through all 393 accessions by the HaplotypeCaller module of GATK v3.8[74] in GVCF mode with default parameters. Then the joint call was performed using the GenotypeGVCFs in GATK v3.8 on the gvcf files of a subset of 386 accessions on A&B subgenomes and 313 accessions on the D subgenome with default parameters (Supplementary Data 1).

SNPs were preliminarily filtered using the VariantFiltration function in GATK v3.8 with the parameter "-filterExpression QD < 2.0 ‖ FS > 60.0 ‖ MQRankSum < −12.5 ‖ ReadPosRankSum < −8.0 ‖ SOR > 3.0 ‖ MQ < 40.0 ‖ DP > 30 ‖ DP < 3." The filtering settings for INDELs were "QD < 2.0, FS > 200.0," and "ReadPosRankSum < −20.0 ‖ DP > 30 ‖ DP < 3". Those sites and alternative alleles that fail the filtering conditions were removed from the dataset. Finally, the identified SNPs and INDELs were further annotated with SnpEff v4.3[75].

**Modeling the genetic distance distribution.** The genetic distance distribution was characterized with 1000 randomly chosen accession pairs. For each pair of accessions, the average distance of each non-overlapping 5 Mbp genomic window was calculated using PLINK v1.9[76]. Gaussian distributions were fitted based on the genetic distances by Expectation-Maximization (EM) algorithm implemented in the R package mixtools v1.2.0[77] of function normalmixEM with the number of components of 2 (for A&B subgenomes) or 1 (for D genome). The threshold separating two subdistributions in the A&B subgenomes was therefore set to $10^{-3}$ variants/bp.

**Dating based on genomic variant density.** We estimated the divergence time (T) corresponding to the mean and mean±sd (D) of genetic distance for each Gaussian component (Supplementary Table 2) by the formula $T = D/2\mu$. The neutral mutation rate (μ) was set to $1.6 \times 10^{-8}$ $nt^{-1}$ $year^{-1}$ according to Dvorak et al[78].

**The IntroBlocker algorithm.** Step 1, Binwise clustering. For each nonoverlapping genomic window (5 Mbp by default), accessions are clustered with an average-linkage hierarchical clustering algorithm based on the pairwise genetic distance matrix, using the hclust function implemented in R. The size of the window was set to 5 Mb in default, which could be modified according to LD-decay distance in specific species. The initial AHGs were assigned by cutting the tree at the height of $10^{-3}$ variant/bp independently for each window.

Step 2, Global AHG re-assignment. AHGs are re-assigned based on the initial AHGs to minimize the transition number between adjacent bins. The re-assignment uses an accession-priority-based AHG referring strategy. For a given priority order, the accession with the highest priority is assigned with a uniform AHG type. For each accession along with this priority order, if its windows are of the same initial AHG type with a high-priority accession, the AHG type of high-priority accession is assigned to the current accession. Otherwise, a novel AHG type is assigned and dominated by the current accession.

The priority order of accessions used in the AHG referring strategy was determined by a bubble sort-like method consisting of two parts. Step 2.1 aims to select a subset of accessions covering the majority of AHG diversity. We implemented an iteratively greedy heuristic algorithm to select one accession in each round. The accession that shares the most initial AHGs with all other accessions is selected in the first round. In the following rounds, the accession that shares the most uncovered AHGs with unselected accessions is selected one by one until most of the initial AHGs could be found in the panel (95% in default). The remaining accessions are randomly appended to the end of the panel.

Step 2.2 aims to find the order that could minimize the transition number by swapping accessions iteratively based on the primary order from step 2.1. In each iteration, adjacent accessions along the priority order were swapped, AHGs were re-assigned using the current order and the transition number was counted. If the total transition number decreases after swapping, the new order is kept. Otherwise, the original order is maintained. The swapping procedure keeps running until no adjustment could be made.

The re-assignment has two modes (Supplementary Fig. 5). In un-supervised mode, the re-assignment was conducted on all accessions globally. In semi-supervised mode, a pre-defined group could be assigned to each accession based on prior knowledge, and the order of groups should be given, such as the order used in this study "WE > DT > hexaploid wheat". Within each group, the un-supervised mode is activated internally to generate the priority order independently. The final order is generated by concatenating the order of each group.

Step 3, Bayesian smoothing. A Bayesian method was applied to correct potential noisy blocks that the AHG types might be misassigned under the hard threshold by referring to the neighboring windows (Supplementary Fig. 6). For each window of each accession, IntroBlocker first estimated the possibility of misassignment by the ratio between type I error (α) and type II error (β) of the current classification of AHG type. These two types of errors are quantified by Gaussian distributions based on the genetic distance to the representative accession. If type II error (β) is larger than a given time (10 in default) of type I error (α), IntroBlocker marks this classification as correct and no change will be made. Otherwise, IntroBlocker tries to correct the AHG assignment by taking the possibility of all AHG types in flanking 10 windows into consideration. The product of the β value and number of occurrences for each AHG type is calculated, and the window will be re-assigned to the AHG type that have the maximum value.

**Evaluation of the IntroBlocker algorithm.** We developed a pipeline to simulate individual chromosome-wide genotyping data from various number of differentiated source populations. The simulation consisted of the three following successive steps (1) forward in-time simulation of differentiated source populations, and the target population deriving from the admixture of the differentiated source populations; (2) recapitated the rest of genealogical history relevant to the source populations by running a coalescent simulation back through time and (3) sampling of individuals from the target population to generate the genotyping data sets.

For the forward in-time simulation, we relied on the SLiM v3.7[79] to simulate 1000 founder chromosomes (500 individuals) for each of the source population. The target population, consisted of 1000 chromosomes, derived from the admixture of source populations with equal possibility after 30 generations. The simulation ended after 100 generations. The tree-sequence recording[80] function implemented in SLiM was used to track the true ancestry for each site. The pseudo chromosome we simulated was 100 Mb of length, and the recombination rate was set to $10^{-8}$ per site and per generation.

For coalescent simulations, we used the recapitation method implemented in pySLiM v0.700 (https://github.com/tskit-dev/pyslim) and msprime v1.1.0[81] to fill out the coalescent history of differentiated source populations. Source population were assumed to derive from a single ancestral population under a pure-drift model of divergence. The effective source population size were set to 1000 for all source populations. The coalescent time of source population was set to $10^5$ years. Neutral mutations were added to the chromosomes according to the tree with mutation rate of $1.6 \times 10^{-8}$ per site and per generation[78]. The selfing rate was set to 0.95 to imitate the reproduction mode of wheat.

In the third and last step of the simulation, 20 chromosomes (10 individuals) for each of the source population and 100 chromosomes (50 individuals) for the target population were sampled. The VCF formatted file containing the genotypes of all selected individuals were outputted.

To evaluate the performance of the IntroBlocker algorithm based on the simulation data, we performed the IntroBlocker algorithm on this simulated data in semi-supervised mode, giving source population higher priority than the admixed population. We defined the accuracy metric to be the ratio of chromosome where the inferred local ancestries were correct. Individuals with the overall heterozygous ratio >0.1 were excluded due to its inconsistence to the self-pollination nature of wheat. The result showed that the inference accuracy was high along the continuous genomic tracts. The overall accuracy was from 97.8% (two source populations) to 96.5% (five source populations), which demonstrated the accuracy of IntroBlocker in different situations.

To evaluate the performance of IntroBlocker in real dataset, we found a large scale introgression signal reported on chr2A by the $f_d$ statistic[3,22], where introgressions were from free-threshing tetraploids to hexaploid landraces. By using barley and *Ae. tauschii* as the outgroup, we reproduced the same introgression signal by the $f_d$ value around 0.5 in this region. By applying the IntroBlocker, we found two primary ancestry sources among all samples in this region, and free-threshing tetraploids to hexaploid landraces maintained both kinds of ancestry, which explained why the $f_d$ value was not near 1 (the full introgression signal). We further classified free-threshing tetraploids and hexaploid landraces according to their ancestries in this region. The $f_d$ statistic value of group consisting of accessions sharing the same ancestry was near 0, while the group composed of tetraploid and hexaploid wheat with different ancestry near 1 in this region. This consistence between the $f_d$ statistic and IntroBlocker demonstrated the potential of our method to precisely identify the ancestry pattern of each sample.

**AHG-based PCoA and phylogenetic analysis.** The distance between a pair of accessions was calculated as the total number of genomic windows of different AHG types. Distances from all possible pairs of accessions formed a distance matrix. The njs function in R package ape v5.3[82] was used to reconstruct a neighbor-joining phylogenetic tree from the distance matrix. The length of branches smaller than 0 was set to 0. The tree was visualized using the iTOL online tool v6[83]. The principal coordinates analysis was conducted using the cmdscale function in R based on the AHG distance matrix.

**Saturation test of unique number of AHG types.** Subsets of four groups with sizes from 1 to each group's total number of accessions were randomly sampled 100 times, respectively. The mean, 5th and 95th percentiles of the unique number of AHGs at each size were calculated over all 5 Mbp windows across the A&B subgenomes.

**Structure transitions among the gene pools of polyploid wheat**. The total AHGs from all 5 Mbp genomic windows of all accessions formed the pooled AHGs. We classified the pooled AHGs according to their presence/absence in the four taxonomic groups. The relative size of the gene pool of polyploid wheats and the ratio of transitions among them were evaluated by the number of unique AHG types in A&B subgenomes.

**Genomic contribution of wild emmer to domesticated wheat**. The contribution of WE accessions to domesticated wheat was calculated as the percentage of windows in A&B subgenomes of domesticated wheat that shares AHG with each WE accession. The contribution can be estimated with the following equation:

$$\text{Contribution}_i = \frac{\sum_{j=1}^{N_\text{accessions}} n_{ij}}{N_\text{windows} \times N_\text{accessions}} \quad (1)$$

$n_{ij}$ refers to the number of windows in the $j$th domesticated wheat accession with AHG type that could be found in the $i$th WE accession. $N_\text{windows}$ refers to the total number of non-overlapping 5 Mbp windows. $N_\text{accessions}$ refers to the total number of domesticated wheat. The coverage of each WE accession to domesticated wheat was defined as the percentage of windows with AHG type that could be found in domesticated wheat, which could be calculated as:

$$\text{Coverage}_i = \frac{m_i}{N_\text{windows}} \quad (2)$$

$m_i$ refers to the number of windows in the $i$th WE accession with AHG type that could be found in domesticated wheat. In the calculation of the cumulative contribution of WE accession to domesticated wheat, it was attributed to the accession with higher priority when AHG was shared by more than one WE accession.

**Variant identification and phylogeny construction on chloroplast genome**. The clean reads of 306 accessions (Supplementary Data 4), including 158 tetraploid, 146 hexaploid and 2 *Ae. longissima* were mapped to a modified version of reference genome that combined the sequences of 21 chromosomes from CS reference genome IWGSC RefSeq v1.0[32] with the sequences of chloroplast and mitochondria from TGACv1[84]. The trimming, mapping and calling steps were the same as described in section "Genomic variant calling and filtering". Then the joint call was performed using GATK GenotypeGVCFs for all 306 accessions on the chloroplast genome. SNPs and INDELs were further filtered using Bcftools[85] with the parameter "-i 'MAC > =2'—min-ac=1 -m2 -M2". Finally, a total of 402 high-confidence variants were identified (Supplementary Data 4).

The NJ phylogenetic tree was obtained by calculating the pairwise genetic distances using PLINK v1.9[76] with parameters "–distance square 1-ibs". The tree was constructed using njs method in the R package ape v5.3[82]. The NJ-tree was rooted with two *Ae. longissima* accessions as outgroup.

**Selection signature detection by Shannon diversity index**. We quantified Shannon diversity index (H)[86] difference of AHGs across A&B genomes between taxonomic groups as proxies for selection signatures during evolution. The Shannon diversity index for the $j$th window is calculated as:

$$H_j = -\sum_i^N P_i \times \log_2 p_i \quad (3)$$

where $p_i$ is the frequency of the $i$th AHG type of the out of N total types in the $j$th window. Copy number variation enriched block (CEB) were excluded. Signatures of selection were detected for windows with $H_\text{WE}-H_\text{DT} > 1.5$, $H_\text{DT}-H_\text{LR} > 0.6$ and $H_\text{LR}-H_\text{CV} > 0.3$ in three stages respectively. Similarly, signatures of fixation ($H < 0.05$) were detected for windows for all four groups. Known domesticated and adaptation related genes residing in windows with selection or fixation signatures were considered as candidate genes.

**CNV block and CNV enriched block (CEB) detection**. Copy number variation (CNV) blocks were defined by a read depth based method for each accession. The read depth (DP) was calculated for all non-overlapping 5 Mbp windows using bedtools v2.27.1[87], and normalized by the average coverage of the A&B subgenomes. Windows with DP < 0.5 or DP > 1.5 were defined as CNV block. To reflect the relative difference in frequency of CNV among groups, we calculated Max$\Delta$(CNV) for each window as:

$$\text{Max}\,\Delta(\text{CNV}) = \max(|F_\text{WE} - F_\text{DT}|, |F_\text{DT} - F_\text{LR}|, |F_\text{LR} - F_\text{CV}|) \quad (4)$$

F refers to the frequency of CNV blocks in each of the four groups. A total of 46 windows were detected as CEBs with Max$\Delta$(CNV) > 0.4.

**centAHG positioning**. We used a sliding-window based method on transition number to position centromeric AHG (centAHG). First, transition number between all pairs of successive windows were counted, and the median of transition number among flanking 16 windows were re-assigned to each window to smooth potential noises. The first of five continuous windows with transition number under 15 was detected as the beginning boundaries of centAHG. Similarly, the last of five continuous windows with transition number under 15 was detected as the

ending boundaries of centAHG. Finally, apparent mis-positionings were corrected through manual consolidation.

**centAHG typing**. Recombination among AHG scarcely happened within the boundaries of centAHG, the major AHG type within these regions was assigned as the type of centAHG for each accession of A&B subgenomes. To this end, we hierarchically clustered centAHGs of all accession using the number of differences of AHG type within the boundaries of centAHG as the distance for pair of accessions for each chromosome. The centAHG types were assigned by cutting the tree at the relative scale of 0.5.

**centAHG-based phylogeny**. The distance between a pair of accessions was defined as the total number of differences in centAHG type. Distance from all possible pairs of accessions formed the distance matrix. The njs function in R package ape v5.3[82] was used to reconstruct a neighbor-joining phylogenetic tree from the distance matrix. The length of branches smaller than 0 were set to 0. The tree was visualized using the iTOL online tool v6[83]. CV accessions were excluded due to the possible shuffling of centAHGs during breeding programs.

**3B chromosome centAHG dating**. Accessions with three major centAHG-3B types were clustered into three groups. Within each group, genetic distances of all windows within the boundaries of 3B centAHG were calculated for all possible pairs. Windows with AHG types that mismatched the corresponding centAHG type were excluded. The genetic distance peaks for 3B-Cent-H1, 3B-Cent-H2 and 3B-Cent-H3 were $2.9 \times 10^{-4}$, $2.2 \times 10^{-4}$, and $1.3 \times 10^{-4}$ variants/bp respectively. The corresponding times were estimated as ~9200, ~6900 and ~4100 years ago, with a neutral mutation rate (μ) of $1.6 \times 10^{-8}$ nt$^{-1}$ year$^{-1}$ as mentioned previously.

**Geographical distribution and spreading route reconstruction**. Genotype data in VCF format of additional 281 landrace accessions were collected from four sources[3,5,18,48]. For each dataset, we merged it with our genotype dataset by preserving variants present in both datasets within the boundaries of centAHG-3B. To determine the type of centAHG-3B, the similarity between each additional accession and the representative accession of 8 major 3B centAHG types (≥5 accessions) was calculated as the percentage of variants identical. The centAHG was assigned as the same to the accession with highest similarity, when the similarity to 1$^\text{st}$ accession is 10% higher than to the 2$^\text{nd}$ accession. Finally, the centAHG-3B types of 240 accessions were among the three major types. Only landraces located in the Eurasian continent were included in the geographical analysis. A total of 412 accessions were used to explore the geographical distribution of three major 3B centAHG types.

To infer the potential spreading route of three major 3B centAHG types into China, we first clustered accessions in our dataset according to the geographical proximity. Geographical locations with less than 2 belonging accessions were excluded. Within accessions preserving each of the three major centAHG-3B types, average genetic distances of all windows within the boundaries of centAHG-3B for all possible inter-location accession pairs formed the distance matrix. Undirected edges representing minimum total phylogenetic length among locations were inferred by the minimum spanning tree algorithm implemented in the mst function from R package ape v5.3[82] and were projected on the map. The directions of routes and the correspondences to historical spreading route were determined by prior assumptions from the literature[14].

**Reporting summary**. Further information on research design is available in the Nature Research Reporting Summary linked to this article.

## Data availability

The raw sequence data generated in this study have been deposited in the Sequence Read Archive under accession code PRJNA759292. The raw sequence data generated in this study have also been deposited in the Genome Sequence Archive under accession code PRJCA006360. The raw sequence data of previously published re-sequenced accessions used in this study are available in the Sequence Read Archive under accession code PRJNA476679, PRJNA596843, PRJNA544491, PRJNA492239, PRJNA528431, PRJNA544491, PRJNA439156, PRJNA663409 and PRJEB22687. The raw sequence data of previously published re-sequenced accessions used in this study are available in the Genome Sequence Archive under accession number CRA003763 and CRA001873. The whole-exome capture dataset is available at http://wheatgenomics.plantpath.ksu.edu/1000EC/. The Chinese Spring wheat reference genome (IWGSC RefSeq v1.0) is publicly available at [https://wheat-urgi.versailles.inra.fr/Seq-Repository/Assemblies]. Source data are provided with this paper.

## Code availability

The ancestry inference software IntroBlocker is available at GitHub [https://wangzihell.github.io/IntroBlocker][88]. The scripts developed in the study have been deposited at GitHub [https://github.com/wangzihell/CAU-MosaicWheat][89].

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

## Acknowledgements

We thank Assaf Distelfeld (Tel Aviv University), Yu Jiang (Northwest A&F University), Xueyong Zhang (Institute of Crop Sciences, Chinese Academy of Agricultural Sciences), Fei Lu and Yuling Jiao (Institute of Genetics and Developmental Biology, Chinese Academy of Sciences) for helping access resequencing data of tetraploid wheat and hexaploid wheat. We thank Qingxin Song (Nanjing Agricultural University) for helping access the data related to the A/B compartment of the wheat genome. We thank Zhenshan Liu (Northwest A&F University), Yongming Chen and Yujiao Gao for their valuable suggestions and comments. This project is supported by the Major Program of the National Natural Science Foundation of China (No. 31991210) to Q.S. This work is also supported by Frontiers Science Center for Molecular Design Breeding (No. 2022TC152), the 2115 Talent Development Program of China Agricultural University, and Hainan Yazhou Bay Seed Lab (No. B21HJ0001) to W.G.

## Author contributions

Q.S. and W.G. supervised the study. Q.S., W.G., and Z.N. conceived this work. Z.W., W.W. and W.G. implemented the algorithm. Z.W. and X.X. collected the raw sequencing data and performed alignment. Z.W., W.W. and X.X. carried out the analysis. Z.W., Y.W., Z.Y., H.P., M.X., Y.Y., Z.H., J.L., Z.S., C.X., B.L., Z.N., W.G., Q.S. interpretated the results. Z.W. and W.G. wrote the manuscript and Q.S. revised it. All authors discussed the results and commented on the manuscript.

## Competing interests

The authors declare no competing interests.
