## [Peer Review File · Nature Communications]

Dispersed emergence and protracted domestication of polyploid wheat uncovered by mosaic ancestral haploblock inferenceReviewers' Comments:

Reviewer #1:

Remarks to the Author:

Nested origin and protracted domestication process of tetraploid wheat uncovered by mosaic ancestry block dissection.

Wang and colleagues build on a suite of papers that have emerged in recent years exploring the possibility of introgression between wheats of different ploidies in their domestication history. Before commenting on Wang, I have to admit that I have been less than convinced by a large number of aspects of these other publications, and consequently some of the basic premises this manuscript inherits I suspect are not well founded. These papers all operate under an assumption (perhaps not unreasonable), that the diversity of the AB genomes observed in hexaploidy wheats but not the D genome must reflect post hexaploidization introgression (ie hexaploids crossing with tetraploids, as opposed to the alternative that hexaploid formation occurred numerous times with a narrow population of D genome donors). Personally, I am agnostic to these two possibilities. Both are biologically possible, but this suite of papers have rather seized on the former with what I consider rather unconvincing evidence but not appropriate to go into here. I am happy to share if required.

Obviously, I concentrate on the Wang manuscript here which produces a potentially useful method for unscrambling the chimeric products of recombination. In general, much of the interesting and potentially useful inferences are towards the end of the paper, tracking blocks through time. There is rather too much emphasis on 'rediscovering' aspects of the process that are very well known, and it does not manuscript the paper well to claim their discovery.

Before dealing with the algorithm, some of the basic premises.

The authors observe an interesting bimodal pattern to variation in tetraploid and hexaploidy wheats. Using molecular clock rate estimates, these appear to coincide with the formation of the domesticate groups and therefore represent subsequent diversification, and then an older peak which the authors attribute to introgression. This would not be my first guess, and makes little sense on the face of it. I would have first assumed the older peak represents the antiquity of the A and B genomes as represented in the diversity present in wild populations. We would of course expect much of this diversity to have become incorporated into the domesticated populations – to assume otherwise is to take a very outdated position that the domesticate populations started out with no diversity at all. This really contrasts with the general field of domestication. Had the author list included students in this field, archaeologists for instance, this kind of erroneous simplifying assumption would not have been made. I don't think this necessarily destroys the analysis, but a substantial rethink and rewrite are in order.

The algorithm itself may be useful, but as it is currently presented is still practically impossible to follow. Focussing on Figure 1, the significance of the use of the colours is not clear, nor is the relative positioning of the cells between step 1 and step 3. I would suggest a more distinctive colour palette for steps 1 and 3, or some clearer indication that these are different colour palettes. I am not sure whether the cells have been shuffled between these two steps. The logic as far as I can follow it seems to suggest that they have, but then the subsequent panels e and f would not be possible, in which case the cartoon flow chart seems baffling in terms of the accessions that have become united by the algorithm in step 3. I'm afraid I found Supplementary Figure 5 of no help at all. I have in the past developed algorithms to unscramble chimera, and I am sure the authors have a sound method, but without clearer explanation it is not yet of use. I am for instance left wondering why accessions 1 and 5 (counting down from the top) in step 1 are not united in step 3 at the right hand side of the grid where they apparently share a clade. Again, a substantial rewrite to clarify this is needed.

The team go on to use NJ algorithms on genome wide data. This is something that the previous suite of papers also did despite this being called out over 20 years ago as a flawed approach if the inferences become cladistic. These are population trees and miss much in terms of numbers of origins etc. PCAs are better in this context by removing the flawed assumption of a bifurcating set of relationships. I'm not sure what this tree adds, since it is unsurprising it recapitulates the result from just using the SNPs. The tree cannot possibly, as the authors state on lines 141-142, demonstrate continuous gene flow from wild emmer, to domesticated tetraploids, to landraces to cultivars.

It is therefore a little difficult to thoroughly review the subsequent findings of the manuscript. However, I would point out that the lack of AHGs found in wild emmer is likely to be a bias in the subjectivity of the method in that wild emmer is given priority in forming AHG groups that are subsequently searched for in more derived forms. It therefore seems likely that you are going to get more complete blocks in this ancestral group – it is not a real representation of the lack of recombination inherent in those wild populations (lines 192-194).

Lines 196-198 : Meanwhile, our results showed that WE accessions have almost equal contributions to the gene pool of DT and HW (Supplementary Fig. 10), suggesting that substantial wild-to-crop introgression occurred in the domestication of tetraploid wheat rather than hexaploid wheat. Why the obsession with introgression here? Surely this is simply the inheritance of genetic diversity through the process of domestication? Again the authors do not seem aware that the general consensus is that large populations contributed to the domesticate populations. The same points apply to the chloroplast tree, in which incidentally, I cannot see any cultivar branches.

Lines 203-205: Hexaploid wheat is thought to emerge from the hybridization between tetraploid *T. turgidum* (AABB) wheat and the diploid species *Aegilops tauschii* (DD)28, while solid genetic evidence is lacking.

No, genetic evidence has been solid for over 3 decades now. The subsequent inference based on the cpDNA investigation:

Notably, we found durum wheat in all five main clades of the hexaploid subgroup (Fig. 3d), implying that the birthplace of hexaploid wheat was likely to be a durum wheat field and that the spread of hexaploid wheat was accompanied by interploidy introgressions in a durum-dominated background. Is an egregious over interpretation and contrasts to the archaeological evidence.

Minor points:

References 3 and 4 appear to be incomplete.

Line 73: the correlation with nucleotide diversity and inferred introgression is not a simple positive one, very high levels of inferred ancestry are associated with lower levels of diversity. Not that I am convinced of the inference of introgression in these cases!

Reviewer #2:

Remarks to the Author:

The article from Wang and colleagues describes a deep analysis of resequencing data of a large panel of domesticated wheat and wild relatives, especially tetraploids, which aimed at reconstructing the complex history of domestication of durum wheat and bread wheat.

The originality of the paper relies on identifying haplotypes along chromosomes in order to dissect ancestral introgression blocks. The main idea is to take advantage of the bimodal distribution of the genetic distances observed when comparing two genotypes. Regions originating from introgressions can be delineated by scanning the level of diversity (i.e. variant density) which increases to $>0.1\%$ (10^{-3}) meaning that segments sharing less than 99.9% between two cultivars could be evidence for

ancestral introgression. Based on this, the authors developed a new method implemented in IntroBlocker which is based on calculating the % divergence in all nonoverlapping 5 Mb windows (ca. 130 windows per chromosome on average). It assigns each window to an ancestral haplotype groups (AHGs) and provides a remarkably clear view of the importance of ancestral introgressions into wheat genome evolution through domestication. This work is of significance to the field of crop genomics and evolution.

The paper is well written and the figures are of high-quality. The results are quite interesting and the methods are sound, as far as I am able to understand!!! I have to admit I am not fully aware of the recent literature on this topic (many things published in the past years about wheat phylogeny, hybridizations, introgressions, etc.) so it is not easy, for me, to clearly identify what is new in terms of wheat reticulated evolution, versus what was previously shown. The wheat scientific community recently entered in a new area, with the availability of ~15 high-quality genome assemblies, which offered the opportunity to address pangenomics questions and analyze the diversity through haplotype reconstruction. The Brinton et al. paper (Comm Biol 2020) was a first step and the current paper appears to me highly original and fully benefits from the haplotype-derived methods to assess the importance of wild-to-crop and interploidy introgressions in the wheat evolution.

I have no major concern about the methods used, the data, the way things were analyzed, nor with the interpretations. However, although it is well written, the paper is extremely hard to understand in details. This is my only major concern. I suggest to try to make it easier to understand. This is true for the text, but this is even more true for the figures that are really challenging. I found there were too many panels, too many details, sometimes not providing information necessary to support the conclusions. Excepted Fig. 6 which is "simple" and a good graphical summary of the evolution model proposed here, the other figures are extremely busy: they represent 32 panels over 5 Figures... With an additional 22 supplementary figures!!! this is plethoric and I have to say I cannot spend enough time to make a review that considers all these details.

SPECIFIC REMARKS:

INTRODUCTION

Introduction is well written and focused on the main message of the paper. However, I found it quite short and I would have appreciated to read more about what was already known. Twenty papers are cited along 2 small paragraphs and it is not possible to really get the state of the art. I suggest to give a bit more details if possible. I especially think about Brinton et al. who previously dissected haplotypes from the wheat assembled genomes.

RESULTS

- + Genomic segments of high variant density attributed to wild-to-crop introgression
- I wondered how the authors discriminated ancestral introgressions (along the domestication process) from recent introgressions selected by breeders?
- The authors refer to Supplemental Fig. 1 to conclude that "few blocks were observed on chromosomes on the D subgenome", but the figure shows only chr4D and only 1 dot (5 Mb window) with an increased diversity level. So, I am not sure I understand the point here.

- + Dissecting the mosaic ancestry introgression blocks via IntroBlocker
- Regarding Fig. 1, I found it really hard to understand. I did not get the priority ranking for instance? What is the purpose of that? I cannot get the sense of Step2 in IntroBlocker as it is explained at lines 111-116. I suggest to try to make the text easier to follow for non-experts by explaining the purpose of each step and why it is important.

- + AHG reveals gene flow from wild emmer to hexaploid wheat
- Fig. 2a is cited here, after the other panels. Not easy to follow the text and figures.
- Fig. 2g panel cited does not exist
- Figure 2 is extremely complicated to understand, with many colors, many small details, which make

it a challenge to fully understand. Panel f is the worth, especially percentages within circles? what do the colors represent here? 1.39% of what? In panel e, what does the 5% threshold represent? a frequency of 5%? but the y label is the AHG frequency? This is confusing. I suggest to explain what is behind each percentage because it is far from obvious to me.

- The results mentioned about "ranked AHGs" (lines 160-170) are again extremely hard to understand. For instance, the sentence "We showed that four groups shared 50.6% of all pooled AHGs, and an additional 22.9% were shared by three groups [...], indicating the continuous transition of gene pools throughout domestication and hexaploidization." I think I understand but it is not obvious to get what do the percentages represent exactly. What is a "group" here? I guess you talk about HW DT etc. but not sure.

I can only suggest to try to make the text easier to follow because the results are interesting but I am afraid it is written exclusively for specialists.

+ Tracing the origin of hexaploid wheat

- Really interesting actually.

- I will not comment every paragraph, but my remarks are the same for the rest of the paper: try to make it easier to follow. I recognize the results are interesting but the difficulty to understand (Fig. 4 for instance... out of my abilities) the details of the results is somehow frustrating.

METHODS

- line 613: the sentence has no verb. To be corrected

Reviewer #3:

Remarks to the Author:

This paper reports a study of the origin of domestication of wheat using genomic data. The authors sequenced 16 new genomes and combined their data with 377 published genomes, and present a new algorithm, IntroBlocker, which is supposed to improve our knowledge on the history of domestication of wheat.

This paper suffers from several weaknesses.

The general structure of the paper is not properly balanced. After a one-page introduction, the results are presented over 11 pages, and then followed by a one-page discussion. The methods are described at the end of the paper over more than 7 pages. This imbalance makes the reading difficult.

There is no clear hypothesis(es), and the review of the current knowledge on wheat domestication is very succinct and does not give a clear picture of the proposed hypotheses to explain the origin of existing wheat varieties.

The proposed method, IntroBlocker, is described in the introduction as a "novel algorithm designed for accurately dissecting ancestral introgression blocks". Several methods in the literature have been published to infer ancestral blocks, including some Bayesian methods to infer admixture and/or introgression. A comparison between the method proposed here and those from the literature is required, as well as an explanation of why the former is an improvement compared to the latter. Unfortunately, no results on the statistical properties of the method proposed by the authors are presented, thus it is impossible to know how it behaves in different situations. Furthermore, IntroBlocker does not seem to be based on a statistical model, so that it is hard to see what it is really tested or measured with this method.

Little details are given on the genomic data analyses. However, there is growing concern that these details have a major impact on the final results such as the identification of SNPs.

Although the grammar and vocabulary are overall correct, the text is sometimes difficult to read because of the many abbreviations used.

The genera should be spelled in full when they appear for the first time in the text.

Reviewer #1

Nested origin and protracted domestication process of tetraploid wheat uncovered by mosaic ancestry block dissection.

1.1 Wang and colleagues build on a suite of papers that have emerged in recent years exploring the possibility of introgression between wheats of different ploidy levels in their domestication history. Before commenting on Wang, I have to admit that I have been less than convinced by a large number of aspects of these other publications, and consequently some of the basic premises this manuscript inherits I suspect are not well founded. These papers all operate under an assumption (perhaps not unreasonable), that the diversity of the AB genomes observed in hexaploid wheats but not the D genome must reflect post hexaploidization introgression (ie hexaploids crossing with tetraploids, as opposed to the alternative that hexaploid formation occurred numerous times with a narrow population of D genome donors). Personally, I am agnostic to these two possibilities. Both are biologically possible, but this suite of papers have rather seized on the former with what I consider rather unconvincing evidence but not appropriate to go into here. I am happy to share if required.

Response:

We agreed that the post-hexaploidization introgression and the multi-time hexaploidization are both possible evolutionary scenarios, and both of them could lead to the uneven distribution of genetic diversities among wheat subgenomes. Our current knowledge of the reticulate evolution among *Triticum* species was far from clear¹. Furthermore, the hypothesis of multi hybridization origin for hexaploid wheat was proposed at times^{2,3}.

One key evidence that supports the post-hexaploidization introgression scenario over the multi hybridization origin was the fact that the natural hybridization between *Triticum turgidum* L and *Aegilops tauschii* was hard to occur⁴, while gene flow between tetraploid and hexaploid wheat happened more frequently¹. *Ae. tauschii* readily hybridizes with tetraploid wheat, and triploid hybrids often produce so many unreduced gametes that they are fertile⁵.

We found that the admixture of ancestries had happened in the domesticated tetraploid wheat populations before the origin of hexaploid wheat and that the admixture process happened along the course of domestication. Our evidence presented was not sufficient to rule out any one of the two possibilities. We think it is likely both these processes contributed to the uneven distribution of genetic diversity. On the other hand, how many times hexaploidization

happened was not our key question to answer in the present work. We focused on the question that how these mosaics of genomic segments of different ancestries contributed to the domestication process of wheat.

1.2 Obviously, I concentrate on the Wang manuscript here which produces a potentially useful method for unscrambling the chimeric products of recombination. In general, much of the interesting and potentially useful inferences are towards the end of the paper, tracking blocks through time. There is rather too much emphasis on ‘rediscovering’ aspects of the process that are very well known, and it does not manuscript the paper well to claim their discovery.

Response:

We are honored that the reviewer found our work of interest. We admitted that there were relatively redundant results in the first version of the manuscript, some of which have no significant advances over those have reported in the literature. To make our manuscript concise and claim our advances well, we removed 4 panels over the first 5 figures (the previous Figure 4a, Figure 5b, Figure 5e, Figure 5f), and reduced the corresponding content in the revised manuscript. We have reduced the content describing the population structure and genetic pool of wheat (most relevant to Figure 2), which has been partially explored by SNP-based approaches. In addition, we substantially improved 9 panels (Figure 1c, 1d, 2a, 2c, 2e, 2f, 3a, 3d, 5b) and enriched sections describing the early admixture of wild lineages, the protracted accumulation process of domestication-related loci, and the AHG blocks in the central zone of the chromosome (centAHG). The details about these revisions have been included in this response letter under the relevant concerns.

1.3 Before dealing with the algorithm, some of the basic premises. The authors observe an interesting bimodal pattern to variation in tetraploid and hexaploidy wheats. Using molecular clock rate estimates, these appear to coincide with the formation of the domesticate groups and therefore represent subsequent diversification, and then an older peak which the authors attribute to introgression. This would not be my first guess, and makes little sense on the face of it. I would have first assumed the older peak represents the antiquity of the A and B genomes as represented in the diversity present in wild populations. We would of course expect much of this diversity to have become incorporated into the domesticated populations – to assume otherwise is to take a very outdated position that the domesticate populations

started out with no diversity at all. This really contrasts with the general field of domestication. Had the author list included students in this field, archaeologists for instance, this kind of erroneous simplifying assumption would not have been made. I don't think this necessarily destroys the analysis, but a substantial rethink and rewrite are in order.

Response:

We fully understand the reviewer's concern. Although we thought that the reviewer's interpretation for the older peak of the genetic diversity was actually similar to ours, that the high diversity genomic blocks came from wild populations. The reason why it seems to be in contrast is the domestication framework under which to interpret the mechanism these diversities integrated into the domestication population, as discussed previously. We do not assume that domestication depleted all diversity. Instead, we found that the admixture of ancestries that had happened before the hexaploidy event. We also noticed that there is emerging evidence suggesting domestication bottleneck had far less influence on the domestication process⁶, and a large proportion of genetic diversity was inherited directly. Although we found that introgression contributed to the diversity recovery in bread wheat, as reported by a suite of recently published large-scale genome sequencing study⁷ and other plant⁸.

We admitted that we over-emphasized the effect of introgression in the previous interpretation of the older peak. Much of these diverse blocks were admixed and incorporated into the hexaploid wheat along the domestication process. The mosaic pattern of ancestry along the genome revealed by the algorithm was not solely resulted from introgression. We have revised Figure 1c to reflect the fact that a large population was involved in the domestication process. We have revised the interpretation of the older peak as follows “Thus, we hypothesized that the mosaic of high-density blocks presented the antiquity of the A and B genomes as represented in the diversity in wild populations, which have become incorporated into the domesticated populations through interploidy introgression or direct inheritance”. In addition, we have added a paragraph discussing the possibility of the two evolutionary scenarios in the discussion section as follows “In contrast to rice and maize, hexaploid wheat emerged in a field of domesticated tetraploid wheat and interploidy hybrid swarms happened in the early agricultural field⁹. This complicated picture of the early domestication process¹ implying the contributions of interploidy and interspecific gene flow to the evolution of Triticum species might be underestimated previously. Recent evidence of

pervasive introgression in wheat^{7,10} also supported that the bread wheat utilized the existing gene pool of tetraploid wheat through interploidy introgression during spreading⁴. However, as it was suggested by the growing archaeogenomic and archaeobotanical evidence from various crops, populations involved in the process of domestication were likely to be large⁶. In addition, the multiple hybridization hypothesis of hexaploid wheat was proposed based on the evidence from specific genomic signatures of *Ae. tauschii* and the D subgenome of bread wheat^{2,3}. Overall, it is likely that both direct inheritance of admixed ancestries through the process of domestication and interploidy introgression contributed to the origin of hexaploid wheat.”

Figure R1: A model for stratified variant density in the A&B subgenomes. The admixture of the interploidy introgression from highly diversified tetraploid wheat (BBAA) introduced high-diversity blocks to hybridized hexaploid wheat (BBAADD), leading to bimodal distribution in A&B subgenomes and unimodal distribution in the D subgenome

1.4 The algorithm itself may be useful, but as it is currently presented is still practically impossible to follow. Focussing on Figure 1, the significance of the use of the colours is not clear, nor is the relative positioning of the cells between step 1 and step 3. I would suggest a more distinctive colour palette for steps 1 and 3, or some clearer indication that these are different colour palettes. I am not sure whether the cells have been shuffled between these two steps. The logic as far as I can follow it seems to suggest that they have, but then the subsequent panels e and f would not be possible, in which case the cartoon flow chart seems baffling in terms of the accessions that have become united by the algorithm in step 3. I'm afraid I found Supplementary Figure 5 of no help at all. I have in the past developed algorithms to unscramble chimera, and I am sure the authors have a sound method, but

without clearer explanation it is not yet of use. I am for instance left wondering why accessions 1 and 5 (counting down from the top) in step 1 are not united in step 3 at the right hand side of the grid where they apparently share a clade. Again, a substantial rewrite to clarify this is needed.

Response:

Thanks for the reviewer's valuable suggestions for visualizing the procedures of the algorithm. We have improved Figure 1d, the flowchart of IntroBlocker algorithm accordingly. The revision includes the following three aspects. (1) We replaced the colour palettes used for mosaic genomic blocks in steps 1 and 3 with two sets of more distinctive ones. In this revised figure, the first palette was used to represent the initial clustering process in step 1. The second palette was used in both mosaic blocks in step 3 and the newly added circles as the representatives of accessions. In addition, nonessential colors previously used to indicate the genomic position were removed. (2) To reflect the change of sample order in the algorithm, we added a column of colored circles along with the mosaic genomic blocks as the representative of accessions. (3) To make the union process of genomic blocks more intuitive, we reduce the number of genomic blocks involved and simplified the example used in the diagram.

In the improved flowchart, we could see that the order of samples (rows of genomic blocks) has been changed in step 2, while no shuffling of columns of genomic blocks was involved. We have substantially rewritten the description about step 2 of the IntroBlocker algorithm as following "Based on the premise that the ancestry of adjacent genomic regions tended to be the same, in step2 IntroBlocker tried to find a priority ranking of accessions, which could minimize the time of AHGs transitions between adjacent windows using the priority-based strategy implemented in step 3. This priority order was created either through a *de novo* ranking algorithm or by evoking the ranking algorithm within each manual assigning accession group based on prior knowledge (**Supplementary Fig. 5**)."

Figure R2: Schematic overview of the IntroBlocker algorithm to infer the AHG types. First, for each 5 Mbp window, the samples hierarchically clustered by genetic distance were grouped with a cut-off at the threshold (10^{-3} variants per bp) imputed from a bimodal distribution. Second, the sample priority order was determined by a *de novo* ranking algorithm aiming to minimize transitions of AHGs (unsupervised) or by ranking within manually assigning groups based on prior knowledge (semi-supervised). Third, AHG types were threaded in a priority-based manner. Finally, a Bayesian approach was introduced for smoothing noisy signals by correcting misassigned blocks based on AHG types of neighboring windows

1.5 The team go on to use NJ algorithms on genome-wide data. This is something that the previous suite of papers also did despite this being called out over 20 years ago as a flawed approach if the inferences become cladistic. These are population trees and miss much in terms of numbers of origins etc. PCAs are better in this context by removing the flawed assumption of a bifurcating set of relationships. I'm not sure what this tree adds, since it is unsurprising it recapitulates the result from just using the SNPs. The tree cannot possibly, as the authors state on lines 141-142, demonstrate continuous gene flow from wild emmer, to domesticated tetraploids, to landraces to cultivars.

Response:

We greatly appreciate your valuable suggestion that PCA fit better to the highly reticulated evolutionary scenario than the NJ algorithms. Accordingly, we replaced the NJ-based phylogenetic tree with PCoA, constructed with AHG-based distances for the A&B subgenomes. One of the major reasons we chose the NJ algorithm to describe the population

structure was that our AHG-based tree recapitulated the SNP-based phylogeny of wheat from a population scale whole-genome resequencing study⁷. From this consistency, we could conclude that the differences in ancestral blocks were a major source of genetic differentiation among samples. We have revised the manuscript as follows “The population structure derived from AHG-based distances resembled the SNP-based phylogenetic relationship for tetraploid and hexaploid wheat⁷, suggesting that the differences in ancestral blocks were a major source of genetic differentiation among samples (Fig. 2a, Supplementary Fig. 12).”

Figure R3, Principal coordinates analysis plot constructed with AHG-based distances for all 5-Mb windows in A&B genomes.

1.6 It is therefore a little difficult to thoroughly review the subsequent findings of the manuscript. However, I would point out that the lack of AHGs found in wild emmer is likely to be a bias in the subjectivity of the method in that wild emmer is given priority in forming

AHG groups that are subsequently searched for in more derived forms. It therefore seems likely that you are going to get more complete blocks in this ancestral group – it is not a real representation of the lack of recombination inherent in those wild populations (lines 192-194).

Response:

Thanks for pointing out this putative drawback of our analysis. To prove that the domesticated tetraploid accessions presented a highly mosaic pattern of AHGs, ruling out the bias bought by priority order used in the algorithm, we inferred the genome-wide AHG for wild emmer and domesticated tetraploid wheat population independently using unsupervised mode. We summarized the AHG transition rate of each chromosome, and the results showed that the domesticated accessions presented a significantly highly mosaic pattern of AHGs. We replaced the previous Figure 3c with independently inferred results.

Figure R4: AHG transition rate of wild emmer (WE) and domesticated tetraploid (DT) group for each chromosome. The inferences were conducted independently for WE and DT using unsupervised mode.

Figure R5: Mosaic graphs of AHGs for 5 WE and 5 DT accessions across chromosome 1A. The inferences were conducted independently for WE and DT using unsupervised mode.

1.7 Lines 196-198: Meanwhile, our results showed that WE accessions have almost equal contributions to the gene pool of DT and HW (Supplementary Fig. 10), suggesting that substantial wild-to-crop introgression occurred in the domestication of tetraploid wheat rather than hexaploid wheat. Why the obsession with introgression here? Surely this is simply the inheritance of genetic diversity through the process of domestication? Again the authors do not seem aware that the general consensus is that large populations contributed to the domesticate populations. The same points apply to the chloroplast tree, in which incidentally, I cannot see any cultivar branches.

Response:

Thanks for your valuable suggestions. We agreed that large populations contributed to the domesticated populations and that much genetic diversity was inherited through the process of domestication. The key message we trying to convey was that various but limited wild populations contributed to the domesticated population. We have revised the statement as follows “Meanwhile, our results showed that WE accessions have almost equal contributions to the gene pool of DT and hexaploid wheat (Supplementary Fig. 10), suggesting that no substantial wild-to-crop introgression occurred in the hexaploid wheat”. In addition, we have added cultivar accessions to the revised chloroplast tree.

Figure R6: Chloroplast SNP-based NJ-tree provides cytoplasmic evidence for the origin of wheat. The tree is rooted by assigning *Ae. speltoides* (SS) accessions as an outgroup. Main clades are separated by grey dash lines. Durum accessions were specifically marked in light blue, and they were found on almost all branches of hexaploid wheat. Clades containing *T.*

polonicum (Polish wheat, tetraploid) and *T. petropavlovskyi* (Xinjiang wheat, hexaploid) were labeled, consistent with reported genomic introgression among the two populations²².

1.8 Lines 203-205: Hexaploid wheat is thought to emerge from the hybridization between tetraploid *T. turgidum* (AABB) wheat and the diploid species *Aegilops tauschii* (DD)²⁸, while solid genetic evidence is lacking. No, genetic evidence has been solid for over 3 decades now.

Response:

We are sorry for this obvious wrong statement. We revised the statement as follows “Hexaploid wheat is thought to emerge from the hybridization between domesticated tetraploid *T. turgidum* (AABB) wheat and the diploid species *Aegilops tauschii* (DD)¹¹, while solid genetic evidence implying which tetraploid subspecies involved is lacking”.

1.9 The subsequent inference based on the cpDNA investigation: Notably, we found durum wheat in all five main clades of the hexaploid subgroup (Fig. 3d), implying that the birthplace of hexaploid wheat was likely to be a durum wheat field and that the spread of hexaploid wheat was accompanied by interploidy introgressions in a durum-dominated background. Is an egregious over interpretation and contrasts to the archaeological evidence.

Response:

We are sorry for the imprecise statement. As the previous response to the reviewer, we have added cultivar accessions to the cpDNA phylogenetic tree (Figure R5). The eukaryotic organelles contain genes with a non-Mendelian mode of inheritance, which makes the chloroplast genome a good system to infer the evolutionary relationships among populations. What we found in this cpDNA phylogenetic tree was that durum wheat was found in all main clades of the hexaploid subgroup, both before or after adding hexaploid cultivar accessions. Together with the evidence from the nucleus genome, we concluded that the durum wheat was likely to play a more important role in the origin of hexaploid wheat than other subspecies of domesticated tetraploid wheat.

This conclusion was drawn in the context of literature, and we were not the first to propose that the durum wheat was the potential female progenitor of bread wheat¹². This hypothesis was supported by multiproxy evidence. In the aspect of time, some of the oldest archaeological remains of free-threshing hexaploid wheat date from 9,700–8,600 cal BP^{11,13},

while the archaeological antiquity of durum wheat is reported to be 10,000 cal BP¹⁴. The durum wheat remains do not preclude the possibility that durum wheat came into existence before the emergence of bread wheat. In contrast, several other species of *T. turgidum*, such as *polonicum*, *turanicum*, and *turgidum*, are probably of relatively recent origin¹⁵. In the aspect of geographical range, Kihara et al.¹⁶ reported the observation that in Ghoznavi, a village near Kosh-Yeylagh, *Ae. tauschii* grew as a weed in a durum wheat field. Until recently, durum wheat cultivation associated with weedy *Ae. tauschii* was observed in the central Alborz Mountain region near northern Iran¹⁷. In addition, a cytological study showed that the durum wheat could be crossed with *Ae. tauschii* and have highly fertile F₁ hybrids¹². While we could not preclude the possibility of other forms of tetraploid Triticum wheat having been involved in the evolution of bread wheat. We have further revised the statement as follows “Notably, we found durum wheat in all main clades of the hexaploid subgroup (Fig. 3d), implying that the durum wheat was likely to play a more important role in the origin and further spreading process of hexaploid wheat among the subspecies of domesticated tetraploid wheat.”

Minor points:

1.10 References 3 and 4 appear to be incomplete.

Response:

Thanks for pointing out this mistake. We have completed these two reference citations.

1.11 Line 73: the correlation with nucleotide diversity and inferred introgression is not a simple positive one, very high levels of inferred ancestry are associated with lower levels of diversity. Not that I am convinced of the inference of introgression in these cases!

Response:

We noticed this result in cases of the cited studies⁷. There existed some regions where high levels of introgression were detected by the f_d statistic, while their genetic diversity (π) might be low. We thought that it was likely to be caused by the inconsistency between the assumed phylogeny and the local ancestry, while IntroBlocker did not have this assumption. We demonstrated that high levels of inferred ancestry inferred by IntroBlocker were associated with high levels of diversity.

Figure R7: Density distribution showing the relationship between nucleotide diversity (π) and the number of AHG types for each 5Mb genomic window.

Reviewer #2 (Remarks to the Author):

The article from Wang and colleagues describes a deep analysis of resequencing data of a large panel of domesticated wheat and wild relatives, especially tetraploids, which aimed at reconstructing the complex history of domestication of durum wheat and bread wheat.

The originality of the paper relies on identifying haplotypes along chromosomes in order to dissect ancestral introgression blocks. The main idea is to take advantage of the bimodal distribution of the genetic distances observed when comparing two genotypes. Regions originating from introgressions can be delineated by scanning the level of diversity (i.e. variant density) which increases to $>0.1\%$ (10^{-3}) meaning that segments sharing less than 99.9% between two cultivars could be evidence for ancestral introgression. Based on this, the authors developed a new method implemented in IntroBlocker which is based on calculating the % divergence in all nonoverlapping 5 Mb windows (ca. 130 windows per chromosome on average). It assigns each window to an ancestral haplotype groups (AHGs) and provides a remarkably clear view of the importance of ancestral introgressions into wheat genome evolution through domestication. This work is of significance to the field of crop genomics and evolution.

Response:

We were honored that the review thought our work to be significant.

2.1 The paper is well written and the figures are of high-quality. The results are quite interesting and the methods are sound, as far as I am able to understand!!! I have to admit I am not fully aware of the recent literature on this topic (many things published in the past years about wheat phylogeny, hybridizations, introgressions, etc.) so it is not easy, for me, to clearly identify what is new in terms of wheat reticulated evolution, versus what was previously shown. The wheat scientific community recently entered in a new area, with the availability of ~15 high-quality genome assemblies, which offered the opportunity to address pangenomics questions and analyze the diversity through haplotype reconstruction. The Brinton et al. paper (Comm Biol 2020) was a first step and the current paper appears to me highly original and fully benefits from the haplotype-derived methods to assess the importance of wild-to-crop and interploidy introgressions in the wheat evolution.

Response:

We are encouraged the reviewer found our work to be interesting. We fully understand the reviewer's concern that the oversimplified introduction might hinder our advances from being well claimed. We have substantially revised this section, focusing on the two following aspects. (1) A more detailed review of research progress relating to the reticulated evolution of wheat as follows "During this domestication process of polyploid wheat, interploidy hybridization is supposed to be involved in the origins of at least two subspecies of *T. turgidum* and two subspecies of *T. aestivum*¹. The diversity level of ancestries was high even within subspecies, as showed by the multiregional origin hypothesis of the domesticated emmer wheat^{18,19}. Population-scale whole-genome sequencing studies have shown that the tetraploid donors contribute more than 20% of the genome in most hexaploid cultivars⁷. Introgressions from more distant species might introduce structural variation^{20,21}, some of which were of high agronomic value²²" and (2) strategies and methods utilized recently to unscramble this complex phylogenetic relationship, including the genome assemblies based studies as follows "Ancestries admixture detection is difficult for species of complex evolutionary history, for which three main categories of methods have been developed. The divergence-based method, such as RND_{min} ²³ and G_{min} ²⁴, can be calculated for individual locus, while they were unsuitable for complex evolutionary scenarios. The phylogenetic relationship-based methods, like f_d ²⁵, benefited from the prior biological knowledge, but their

premise that the populations used as outgroup and reference have uniform ancestry might be over-simplified. The ancestry deconvolution methods, like HAPMIX²⁶, could deal with multiple-way admixture by evaluating ancestry across admixed genomes based on statistical models. But the data type they require for input was more specific, as phased haplotypes and/or reference panels are often required. Furthermore, coalescent simulations of demographic models could be used to estimate introgression rate as a component of broader demographic history, though they tended to report inaccurate estimation of ancestral state when the demographic history of species is relatively complex²⁷. The recently published genome assemblies²⁸ offered the opportunity to address questions through genome alignment strategy²⁹, though the lack of sufficient data hindered their application at the population scale to date. To fill the gaps between the complex genomic architectures and the unresolved domestication process of polyploid wheat, a method that could dissect ancestry genomic blocks utilizing the massive genome-wide sequencing data available is needed.”. We hoped this revision could summarize the recent progress properly.

2.2 I have no major concern about the methods used, the data, the way things were analyzed, nor with the interpretations. However, although it is well written, the paper is extremely hard to understand in details. This is my only major concern. I suggest to try to make it easier to understand. This is true for the text, but this is even more true for the figures that are really challenging. I found there were too many panels, too many details, sometimes not providing information necessary to support the conclusions. Excepted Fig. 6 which is "simple" and a good graphical summary of the evolution model proposed here, the other figures are extremely busy: they represent 32 panels over 5 Figures... With an additional 22 supplementary figures!!! this is plethoric and I have to say I cannot spend enough time to make a review that considers all these details.

Response:

We are sorry for the over-complicated presentation and have reduced the unessential details in the revised manuscript and figures. Overall, we removed 4 panels over the first 5 figures (the previous Figure 4a, Figure 5b, Figure 5e, Figure 5f), and reduced the corresponding content in the revised manuscript. We have reduced the content describing the population structure and genetic pool of wheat (most relevant to Figure 2), which has been partially explored by SNP-based approaches. In addition, we substantially improved 9 panels (Figure

1c, 1d, 2a, 2c, 2e, 2f, 3a, 3d, 5b). The details of these revisions were presented under the relevant concerns. We hoped this revision could make the key message of the paper clearer.

SPECIFIC REMARKS:

INTRODUCTION

2.3 Introduction is well written and focused on the main message of the paper. However, I found it quite short and I would have appreciated to read more about what was already known. Twenty papers are cited along 2 small paragraphs and it is not possible to really get the state of the art. I suggest to give a bit more details if possible. I especially think about Brinton et al. who previously dissected haplotypes from the wheat assembled genomes.

Response:

We fully understood the reviewer's concern that the oversimplified introduction might hinder our advances from being well claimed. We have substantially revised this section as mentioned above, focusing on the two following aspects. (1) A more detailed review of research progress relating to the reticulated evolution of wheat and (2) strategies and methods utilized recently to unscramble this complex phylogenetic relationship, including the genome assemblies-based studies. The details of the revision mentioned here have been included above.

RESULTS

2.4 + Genomic segments of high variant density attributed to wild-to-crop introgression - I wondered how the authors discriminated ancestral introgressions (along the domestication process) from recent introgressions selected by breeders?

Response:

The evolutionary history of wheat was highly reticulated over a long time course, and the introgression events were highly nested. By calculating the genetic distance between AHG in hexaploid wheat and the tetraploid wheat of the same AHG type, we show that the introgression events happened continuously. We admitted that our method could not discriminate between these two types of introgression clearly. In general, the lower differential level an introgression segment was between the donor and the receiver, the higher possibility this segment was selected by breeders recently.

Figure R8: Density distribution of putative diversification time of AHG in hexaploid from tetraploid wheat.

2.5 - The authors refer to Supplemental Fig. 1 to conclude that "few blocks were observed on chromosomes on the D subgenome", but the figure shows only chr4D and only 1 dot (5 Mb window) with an increased diversity level. So, I am not sure I understand the point here.

Response:

We are sorry for this confusing statement. The message we tried to convey is that the chromosomes in the D subgenome have far fewer high-diversity genomic blocks, compared with the that in A&B subgenome. This pattern was consistent for both landrace and cultivar accessions. To show this pattern clearer, we extended Supplemental Fig. 1 to include all D chromosomes. We have revised the sentence as follows "few high-diversity blocks (genetic distance above the inferred threshold) were observed on chromosomes in the D subgenome".

Figure R9: The binwise densities of pairwise genetic differences along 7 D chromosomes for the three accession pairs.

2.6 + Dissecting the mosaic ancestry introgression blocks via IntroBlocker

- Regarding Fig. 1, I found it really hard to understand. I did not get the priority ranking for instance? What is the purpose of that? I cannot get the sense of Step2 in IntroBlocker as it is explained at lines 111-116. I suggest to try to make the text easier to follow for non-experts by explaining the purpose of each step and why it is important.

Response:

We are sorry for the confusing description and flowchart. The purpose of step 2 was to find a priority order of accessions that could minimize the time of AHG transitions between adjacent windows, making the genomic blocks with the same type of AHG as continuous as possible. This was achieved by applying the greedy search strategy and a bubble sort-like algorithm consecutively, as shown by Supplementary Fig. 5. This is also the step where prior biological knowledge could be taken into account (the semi-supervised mode where samples could be assigned into groups).

We have revised Figure 1d, as shown in the response to Reviewer #1. One of the major improvements to this figure is that we demonstrated the change of sample order in the algorithm explicitly, by adding a column of colored circles along with the mosaic genomic blocks as the representative of the sample. We have substantially rewritten the description about step 2 of the IntroBlocker algorithm as following “Based on the premise that the

ancestry of adjacent genomic regions tended to be the same, in step2 IntroBlocker tried to find a priority ranking of accessions, which could minimize the time of AHGs transitions between adjacent windows using the priority-based strategy implemented in the next step. This order was created either through a *de novo* ranking algorithm or by evoking the ranking algorithm within each manual assigning accession group based on prior knowledge (**Supplementary Fig. 5**).”

2.7 + AHG reveals gene flow from wild emmer to hexaploid wheat

- Fig. 2a is cited here, after the other panels. Not easy to follow the text and figures.

Response:

We are sorry for the wrong citation of figures in the manuscript. In line 129, the “Fig. 2e” should be Fig. 1e. In line 133, the “Fig. 2f” should be “Fig. 1f”. We have corrected them in the revised manuscript.

2.8 - Fig. 2g panel cited does not exist

Response:

We are sorry for this apparent mistake. Fig. 2g panel previously referred to the insect panel of Fig. 2f. We neglected to mark the character “g” in the previous version of Figure 2. In the revised figures, the g panel has been removed due to the confusing representation.

2.9 - Figure 2 is extremely complicated to understand, with many colors, many small details, which make it a challenge to fully understand. Panel f is the worth, especially percentages within circles? what do the colors represent here? 1.39% of what? In panel e, what does the 5% threshold represent? a frequency of 5%? but the y label is the AHG frequency? This is confusing. I suggest to explain what is behind each percentage because it is far from obvious to me.

Response:

We are sorry for the confusing visualization and description. We have improved Figure 2e and Figure 2f. Figure 2e intended to show that 2~5 common AHGs (detection frequency $\geq 5\%$) contributed about half of the total AHG across each chromosome in the A&B subgenome. The frequency of 5% was the threshold that partitioned common and rare AHG, while the y

axis was the cumulative frequency of AHG. The improvement of Figure 2e includes the removal of rare AHGs (detection frequency <5%), and the replacement of the y label to “Cumulative frequency”. We have revised the statement about Figure 2e as follows “**The cumulative frequency distribution showed that 2-6 common AHGs (detection frequency $\geq 5\%$) contributed nearly half (48.1%) of each chromosome in the A&B subgenomes (Fig. 2d).**”

We have re-designed Figure 2f, which is intended to show the relative proportion of AHGs with various inheritance patterns. The new Figure 2f used the alluvial format to show the inheritance pattern among the four groups (WE, DT, LR, CV), along with a pie chart showing the relative proportions. We have revised the statement about Figure 2e as follows “**We showed that four groups shared 50.6% of all AHGs, and an additional 21.7% were shared by three consecutive groups (Fig. 2f, Supplementary Table 6), indicating the continuous transition of gene pools throughout domestication and hexaploidization. A total of 19.6% of AHGs were shared by DT, LR, CV, in contrast with 1.1% shared by WE, LR, and CV, supporting DT as the gene pool that directly contributed AHG introgression into the gene pool in hexaploid wheat.**” We wish these revisions could make the message of the figures clearer.

Figure R10, Stacked bar graph showing the cumulative frequency and corresponding order of common AHGs (frequency > 5%) by chromosome. “D” indicates the average value of 7 D chromosomes.

Figure R11, Proportion of shared AHGs among four taxonomic groups shown by alluvial plot revealed genetic flows through the route of WE-DT-LR-CV. A dramatically higher proportion of AHGs in the hexaploid groups can be traced back to the DT group than to the WE group using the collected samples, indicating DT as the closest gene pool for hexaploid wheat. The proportion of AHGs with each inheritance pattern, compared to the total AHGs, was shown by the pie chart.

2.10 - The results mentioned about "ranked AHGs" (lines 160-170) are again extremely hard to understand. For instance, the sentence "We showed that four groups shared 50.6% of all pooled AHGs, and an additional 22.9% were shared by three groups [...], indicating the continuous transition of gene pools throughout domestication and hexaploidization." I think I understand but it is not obvious to get what do the percentages represent exactly. What is a "group" here? I guess you talk about HW DT etc. but not sure.

I can only suggest to try to make the text easier to follow because the results are interesting but I am afraid it is written exclusively for specialists.

Response:

We are sorry for the confusing visualization and description. The “top-ranked AHGs” refer to the AHG that were commonly detected (frequency>5% in the dataset). The key message we want to convey through this metric was that 2~6 common AHGs (detection frequency $\geq 5\%$) contributed about half (48.1%) of the total AHG across each chromosome in A&B subgenome. This part of the result refers to Figure 2e and we have revised the statement as follows “The cumulative frequency distribution showed that 2-6 common AHGs (detection frequency $\geq 5\%$) contributed nearly half (48.1%) of each chromosome in the A&B subgenomes (Fig. 2d).”

The two numbers, “50.6%” and “22.9%”, are the percentage of AHG with different inheritance patterns among four groups. The four groups here refer to wild emmer wheat (WE), domesticated tetraploid wheat (DT), hexaploid landrace (LR), and hexaploid cultivar wheat (CV). For example, we showed that there were 50.6% of all AHG shared by four groups, which means that the AHG type of 50.6% of all genomic windows was present in four groups. This part of the result refers to Figure 2f. We have revised the statement about Figure 2e as follows “We showed that four groups shared 50.6% of all AHGs, and an additional 21.7% were shared by three consecutive groups (**Fig. 2f, Supplementary Table 6**), indicating the continuous transition of gene pools throughout domestication and hexaploidization. A total of 19.6% of AHGs were shared by DT, LR, CV, in contrast with 1.1% shared by WE, LR, and CV, supporting DT as the gene pool that directly contributed AHG introgression into the gene pool in hexaploid wheat.”

2.11 + Tracing the origin of hexaploid wheat

- Really interesting actually.

Response:

We are honored that the reviewer found our work of interest.

2.12 - I will not comment every paragraph, but my remarks are the same for the rest of the paper: try to make it easier to follow. I recognize the results are interesting but the difficulty to understand (Fig. 4 for instance... out of my abilities) the details of the results is somehow frustrating.

Response:

We agreed that the key message we tried to convey are distracted by the unessential details. We have refined the main figures and the manuscript. As for Figure 4, we have removed the previous panel “a” and removed the corresponding description about the evaluation of the genetic bottleneck strength. Overall, we removed 4 panels over the first 5 figures (the previous Figure 4a, Figure 5b, Figure 5e, Figure 5f), and reduced the corresponding content in the revised manuscript. We have reduced the content describing the population structure and genetic pool of wheat (most relevant to Figure 2), which has been partially explored by SNP-based approaches. In addition, we substantially improved 9 panels (Figure 1c, 1d, 2a, 2c,

2e, 2f, 3a, 3d, 5b). The details of these revisions were presented under the concerns where they were relevant. We hoped this revision could make the key message of the paper clearer.

METHODS

2.12 - line 613: the sentence has no verb. To be corrected

Response:

We have corrected the sentence mentioned to “Within each group, genetic distances of all windows within the boundaries of 3B centAHG for were calculated for all possible pairs.”

Reviewer #3 (Remarks to the Author):

This paper reports a study of the origin of domestication of wheat using genomic data. The authors sequenced 16 new genomes and combined their data with 377 published genomes, and present a new algorithm, IntroBlocker, which is supposed to improve our knowledge on the history of domestication of wheat.

This paper suffers from several weaknesses.

3.1 The general structure of the paper is not properly balanced. After a one-page introduction, the results are presented over 11 pages, and then followed by a one-page discussion. The methods are described at the end of the paper over more than 7 pages. This imbalance makes the reading difficult.

Response:

We have been fully aware that the unbalanced structure of the previous version manuscript brought unnecessary difficulty for reading. We have substantially revised the manuscript. The revision mainly includes the following three aspects.

(1) We have enriched the introduction section, by adding a more detailed review of the reticulated evolution of wheat and methods utilized recently to unscramble this complex phylogenetic relationship as follows “Ancestries admixture detection is difficult for species of complex evolutionary history, for which three main categories of methods have been developed. The divergence-based method, such as RND_{\min}^{23} and G_{\min}^{24} , can be calculated for individual locus, while they were unsuitable for complex evolutionary scenarios. The phylogenetic relationship-based methods, like f_d^{25} , benefited from the prior biological

knowledge, but their premise that the populations used as outgroup and reference have uniform ancestry might be over-simplified. The ancestry deconvolution methods, like HAPMIX²⁶, could deal with multiple-way admixture by evaluating ancestry across admixed genomes based on statistical models. But the data type they require for input was more specific, as phased haplotypes and/or reference panels are often required. Furthermore, coalescent simulations of demographic models could be used to estimate introgression rate as a component of broader demographic history, though they tended to report inaccurate estimation of ancestral state when the demographic history of species is relatively complex²⁷. The recently published genome assemblies²⁸ offered the opportunity to address questions through genome alignment strategy²⁹, though the lack of sufficient data hindered their application at the population scale to date. To fill the gaps between the complex genomic architectures and the unresolved domestication process of polyploid wheat, a method that could dissect ancestry genomic blocks utilizing the massive genome-wide sequencing data available is needed.” And about the reticulate evolution of wheat as follows “During this domestication process of polyploid wheat, interploidy hybridization is supposed to be involved in the origins of at least two subspecies of *T. turgidum* and two subspecies of *T. aestivum*¹. The diversity level of ancestries was high even within subspecies, as showed by the multiregional origin hypothesis of the domesticated emmer wheat^{18,19}. Population-scale whole-genome sequencing studies have shown that the tetraploid donors contribute more than 20% of the genome in most hexaploid cultivars⁷. Introgressions from more distant species might introduce structural variation^{20,21}, some of which were of high agronomic value²²”

(2) We have tried to make the main results section more concise, by removing what has been reported in the literature and superfluous details. Overall, we removed 4 panels over the first 5 figures (the previous Figure 4a, Figure 5b, Figure 5e, Figure 5f), and reduced the corresponding content in the revised manuscript. We have reduced the content describing the population structure and genetic pool of wheat (most relevant to Figure 2), which has been partially explored by SNP-based approaches. In addition, we substantially improved 9 panels (Figure 1c, 1d, 2a, 2c, 2e, 2f, 3a, 3d, 5b). Details of these revisions were presented under the concerns where they were relevant. We hoped this revision could make the key message of the paper clearer.

(3) In the discussion section, we have added an important topic about the two possible evolutionary scenarios that could lead to the uneven distribution of genetic diversities among

wheat subgenomes: post-hexaploidization introgression and the multi-time hexaploidization as follows “In contrast to rice and maize, hexaploid wheat emerged in a field of domesticated tetraploid wheat and interploidy hybrid swarms happened in the early agricultural field⁹. This complicated picture of the early domestication process¹ implying the contributions of interploidy and interspecific gene flow to the evolution of *Triticum* species might be underestimated previously. Recent evidence of pervasive introgression in wheat^{7,10} also supported that the bread wheat utilized the existing gene pool of tetraploid wheat through interploidy introgression during spreading⁴. However, as it was suggested by the growing archaeogenomic and archaeobotanical evidence from various crops, populations involved in the process of domestication were likely to be large⁶. In addition, the multiple hybridization hypothesis of hexaploid wheat was proposed based on the evidence from specific genomic signatures of *Ae. tauschii* and the D subgenome of bread wheat^{2,3}. Overall, it is likely that both direct inheritance of admixed ancestries through the process of domestication and interploidy introgression contributed to the origin of hexaploid wheat.” We further discussed centromeric ancestral haplotype groups (centAHG) in a broader context as follows “Studies in sunflower³⁰ and cotton³¹ showed that the haplotypes blocks that keep adaptive alleles together contributed to the ecotypic adaptation, the recombination rate of which were often suppressed inversions. The identified centromeric ancestral haplotype groups (centAHG) might play a similar role in wheat without conspicuous structural variants. They stretched dozens or hundreds of megabase pairs in length, containing thousands of genes. They could be stably inherited from tetraploid wheat under selection and might act as the chromosome backbones through domestication. Significant phenotypic differences have been detected between accession with different centromeric haplotype³², and the centromeric region of the 3B chromosome has been reported to lead genetic differentiation in Chinese landraces³³. These evidences suggested that centAHG might be instrumental in the quick adaption to local environments of wheat during spreading. Furthermore, studies in Triticeae suggested that the homologous centromere functional sequences might be related to homolog pairing and recombination. As rare crossovers were detected inter-centAHGs, modern breeding tended to reduce the diversity of centAHGs by selecting the convergent type. It suggested that uniform centAHGs are important for increasing breeding efficiency³²”

3.2 There is no clear hypothesis(es), and the review of the current knowledge on wheat domestication is very succinct and does not give a clear picture of the proposed hypotheses to explain the origin of existing wheat varieties.

Response:

We fully understand the reviewer's concern. As mentioned by the first two reviewers, one of the major reasons that hindered our advances to be well claimed was that current knowledge on wheat domestication and reticulate evolution were not properly reviewed. We have substantially revised the introduction section as mentioned. Based on the current knowledge, and the contrast of the genetic distance distribution we found on the A&B and D subgenome, we proposed the hypothesis that the genomic ancestry composition was various among accession, which was shaped by homoploid hybridization and interploidy introgression. This admixture of ancestry might play a vital role in the domestication of wheat. We have revised the hypothesis mentioned in the manuscript as follows "Thus, we hypothesized that the mosaic of high-density blocks presented the antiquity of the A and B genomes as represented in the diversity in wild populations, which have become incorporated into the domesticated populations through the early admixture among lineages and the following interploidy introgression". Our picture was that polyploid wheat originated from a dispersed geographic range and the domestication-related loci accumulated in a protracted process. We have revised the statement section "A genetic-based model for the origin and domestication process of polyploid wheat" as follows "We propose a refined model for the origin of wheat with multilevel genetic evidence by highlighting the dispersed emergence and protracted domestication of polyploid wheat (Fig. 6). In this scenario, *T. turgidum* (BBAA) diversified into multiple WE lineages over a long time since its origin by polyploidization, and the low frequency of gene flow allowed the accumulation of genetic diversity among the WE lineages scattered at the beginning of agriculture in the Fertile Crescent area. During the Neolithic period (~10,000 years ago), domesticated emmer emerged from the admixture of wild emmer lineages in the Fertile Crescent and introduced the 1st genomic diversity reduction, but wild-to-crop introgression by unconscious mixing between multiple WE lineages in the various regions, created the mosaic ancestral genome and remarkably increased genetic diversity. The restored diversity in domesticated emmer supports its further spread and continuous domestication, which resulted in multiple tetraploid subspecies, such as durum, rivet and polish wheat. Later, natural hybridization between free-threshing tetraploid wheat (BBAA) and wild goatgrass (DD), which was likely to occur in a durum wheat field, created the first hexaploid wheat (BBAADD). Hexaploid wheat was successfully selected and became dominant in the local field, although farmers were unaware of its ploidy. Although hexaploidization introduced the 2nd genomic diversity reduction, the interploidy

introgression promised by the state of hybrid swarm with already-spread domesticated tetraploid wheat introduced ancestral genomic mosaics to hexaploid wheat. The restored genetic diversity of hexaploid wheat supports its spread and further domestication. Then, hexaploid wheat finally displaced tetraploid wheat and was widely grown worldwide. Modern breeding activity introduced the 3rd genomic diversity reduction, while hybridization followed by selection counteracted this by increasing the genetic combinations. In summary, we proposed that the polyploid wheat originated from a dispersed geographic range and the domestication-related loci accumulated in a protracted process, and the coexistence of domesticated tetraploid and hexaploid wheat in history entangled their domestication process and resulted in shared gene pools.”

3.3 The proposed method, IntroBlocker, is described in the introduction as a "novel algorithm designed for accurately dissecting ancestral introgression blocks". Several methods in the literature have been published to infer ancestral blocks, including some Bayesian methods to infer admixture and/or introgression. A comparison between the method proposed here and those from the literature is required, as well as an explanation of why the former is an improvement compared to the latter. Unfortunately, no results on the statistical properties of the method proposed by the authors are presented, thus it is impossible to know how it behaves in different situations. Furthermore, IntroBlocker does not seem to be based on a statistical model, so that it is hard to see what it is really tested or measured with this method.

Response:

We fully understood the reviewer’s concern about the advance and performance of our algorithm. To claim the advances, we have reviewed current methods available for detecting introgression in the introduction section as mentioned above. The current introgression detection methods often required specific types of input data, such as phased haplotypes, genetic maps, phylogeny, reference panels, etc. These data are either hardly available, or involved error-prone processes when inferring in crops with a large genome at population scale. IntroBlocker did not estimate ancestry coefficients as the parameters of a statistical model, while IntroBlocker utilized statistical models as a part of it, such as the Hidden Markov model (HMM). Instead, IntroBlocker tried to discover ancestral structure within the genotype data in a less parametric way, due to the complex nature of crop genome and evolutionary history. The design of the method was based on the evolutionary model we concluded from the patterns of the density distribution of inter-individual genomic variants.

This is also the key feature that supports the dissection of the mosaic ancestries in the wheat genome.

To prove the performance of our algorithm, we have evaluated the accuracy of the IntroBlocker algorithm both in simulated data and real datasets. Both results showed that the performance is robust and accurate to draw current conclusions. We have added the description of the evaluation of the IntroBlocker algorithm in the result section as follows “We validated the reliability of IntroBlocker in both simulated and real dataset. In various admixture scenarios we simulated, IntroBlocker achieved an accuracy of 97% on average (**Supplementary Fig. 5, Supplementary Fig. 5**). We further compared IntroBlocker with the haplotypes detected by genome f_d statistic^{7,25}. The results obtained through the two methods are basically conformable (**Supplementary Fig. 5**). Furthermore, IntroBlocker could leverage prior knowledge of both biological and evolutionary in a flexible manner.” We have added a section entitled “Evaluation of the IntroBlocker algorithm” describing the evaluation process in the method as follows: “We developed a pipeline to simulate individual chromosome-wide genotyping data from a various number of differentiated source populations.”

“The simulation consisted of the three following successive steps (1) forward in-time simulation of differentiated source populations, and the target population deriving from the admixture of the differentiated source populations; (2) recapitated the rest of genealogical history relevant to the source populations by running a coalescent simulation back through time and (3) sampling of individuals from the target population to generate the genotyping data sets.

For the forward in-time simulation, we relied on the SLiM v3.7³⁴ to simulate 1000 founder chromosomes (500 individuals) for each of the source populations. The target population consisted of 1000 chromosomes, derived from the admixture of source populations with equal possibility after 30 generations. The simulation ended after 100 generations. The tree-sequence recording³⁵ function implemented in SLiM was used to track the true ancestry for each site. The pseudo chromosome we simulated was 100Mb in length, and the recombination rate was set to 10^{-8} per site and per generation.”

“For coalescent simulations, we used the recapitation method implemented in pySLiM (<https://github.com/tskit-dev/pyslim>) msprime³⁶ to fill out the coalescent history of differentiated source populations. Source populations were assumed to derive from a single ancestral population under a pure-drift model of divergence. The effective source population size was set to 1000 for all source populations. The coalescent time of the source population

was set to 10^5 years. Neutral mutations were added to the chromosomes according to the tree with a mutation rate of 1.6×10^{-8} per site and per generation³⁷. The selfing rate was set to 0.95 to imitate the reproduction mode of wheat.”

“In the third and last step of the simulation, 20 chromosomes (10 individuals) for each of the source population and 100 chromosomes (50 individuals) for the target population were sampled. The VCF formatted file containing the genotypes of all selected individuals was outputted.”

“To evaluate the performance of the IntroBlocker algorithm based on the simulation data, we performed the IntroBlocker algorithm on this simulated data in semi-supervised mode, giving the source population higher priority than the admixed population. We defined the accuracy metric to be the ratio of the chromosome where the inferred local ancestries were correct. Individuals with the overall heterozygous ratio above 0.8 were excluded due to its inconsistency with the self-pollination nature of wheat. The result showed that the inference accuracy was high along the continuous genomic tracts. The overall accuracy was from 97.8% (two sources population) to 96.5% (five sources population), which demonstrated the accuracy of IntroBlocker in different situations.”

“To evaluate the performance of IntroBlocker in a real dataset, a large-scale introgression signal was reported on chr2A by the f_d statistic^{7,25}, where introgressions were from free-threshing tetraploids to hexaploid landraces. By using barley and *Ae. tauschii* as the outgroup, we reproduced the same introgression signal by the f_d value around 0.5 in this region. By applying the IntroBlocker, we found two primary ancestry sources among all samples in this region, and free-threshing tetraploids to hexaploid landraces maintained both kinds of ancestry, which explained why the f_d value was not near 1 (the full introgression signal). We further classified free-threshing tetraploids and hexaploid landraces according to their ancestries in this region. The f_d statistic value of the group consisting of accessions sharing the same ancestry was near 0, while the group composed of tetraploid and hexaploid wheat with different ancestry near 1 in this region. This consistency between the f_d statistic and IntroBlocker demonstrated the potential of our method to precisely identify the ancestry pattern of each sample.”

Figure R13: Simulated ancestries and corresponding inference using IntroBlocker along a 100Mb length pseudo-chromosome. Two individuals were shown under the scenario of 2~5 ancestry sources.

Figure R14: Inference accuracy of IntroBlocker under the scenario of 2~5 ancestry sources. 50 replications were conducted for each scenario.

Figure R15: A case study of IntroBlocker algorithm superiority in introgression detection. a the four-taxon topology used for modeling introgression in ABBA–BABA tests. b, three strategies to select a subset of FT and LR samples as the p2 and p3 populations. Select all FT and LR samples regardless of their AHG type in the reported introgression region (indicated by shaded area). Blue and red rectangles indicated the two dominant AHG types detected in this region. c, the corresponding f_d distribution of three classifications mentioned in (b) along the 2A chromosome.

3.4 Little details are given on the genomic data analyses. However, there is growing concern that these details have a major impact on the final results such as the identification of SNPs.

Response:

We fully understood the reviewer’s concern. We have complemented the necessary details relating to genomic data analyses in the “Whole-genome sequencing and quality control” and the “Genomic variant calling and quality control” section of methods. The revision was included as follows:

“Whole-genome sequencing and quality control

Genomic DNA was extracted from young leaves of 16 accessions following a standard CTAB protocol³⁸. DNA libraries were constructed by Novogene and sequenced with the Illumina Hiseq Xten PE150 platform with an insert size of approximately 500 bp at an average depth of 5.2 (Supplementary Table 2). For each accession, the quality control for raw reads was conducted using Trimmomatic³⁹. The leading and trailing low-quality bases (below quality 3) were removed. Each read was scanned with a 4-base sliding window, cutting when the average quality per base drops below 15. Reads below the 36 bases long were removed.

Genomic variant calling and quality control

High-quality clean reads were mapped to the CS wheat reference genome (IWGSC RefSeq v1.0)⁴⁰ using BWA-MEM⁴¹ with default parameters. Read pairs with abnormal insert sizes (>10,000 bp or <-10,000 bp or =0 bp) or low mapping qualities (<1) were filtered using

Bamtools v2.4.1⁴². Potential PCR duplicates reads were further removed using the rmdup function in Samtools v1.3.1⁴³. SNPs and INDELs were identified through all 393 accessions by the HaplotypeCaller module of GATK v3.8⁴⁴ in GVCF mode with default parameters. Then the joint call was performed using the GenotypeGVCFs in GATK v3.8 on the gvcf files of a subset of 386 accessions on A&B subgenomes and 313 accessions on the D subgenome with default parameters (Supplementary Table 1).

SNPs were preliminarily filtered using the VariantFiltration function in GATK v3.8 with the parameter “-filterExpression QD < 2.0 || FS > 60.0 || MQRankSum < -12.5 || ReadPosRankSum < -8.0 || SOR > 3.0 || MQ < 40.0 || DP > 30 || DP < 3.” The filtering settings for INDELs were “QD < 2.0, FS > 200.0,” and “ReadPosRankSum < -20.0 || DP > 30 || DP < 3”. Those sites and alternative alleles that fail the filtering conditions were removed from the dataset. Finally, the identified SNPs and INDELs were further annotated with SnpEff v4.3⁴⁵.”

In addition, the codes involved in this study have been deposited into a public depository at [<https://github.com/wangzihell/CAU-MosaicWheat>].

3.5 Although the grammar and vocabulary are overall correct, the text is sometimes difficult to read because of the many abbreviations used.

Response:

We greatly appreciate your suggestion. The following abbreviations expanded in the revised manuscript as they appeared only a few times: hexaploid wheat (HW), domesticated emmer (DE), free-threshing tetraploid wheat (FT), Chinese landraces (CLs), Non-Chinese landraces (NCLs).

3.6 The genera should be spelled in full when they appear for the first time in the text.

Response:

Thanks for your reminder. We have spelled the full name of *Triticum turgidum* (Line 43) and *Aegilops tauschii* (Line 48) in the revised manuscript.

References

1. Zhao, X., Fu, X., Yin, C. & Lu, F. Wheat speciation and adaptation: perspectives from reticulate evolution. *aBIOTECH* **2**, 386-402 (2021).
2. Gaurav, K. *et al.* Population genomic analysis of *Aegilops tauschii* identifies targets for bread wheat improvement. *Nature Biotechnology* (2021).
3. Giles, R.J. & Brown, T.A. GluDy allele variations in *Aegilops tauschii* and *Triticum aestivum*: implications for the origins of hexaploid wheats. *Theor Appl Genet* **112**, 1563-72 (2006).
4. Matsuoka, Y. Evolution of Polyploid Triticum Wheats under Cultivation: The Role of Domestication, Natural Hybridization and Allopolyploid Speciation in their Diversification. *Plant and Cell Physiology* **52**, 750-764 (2011).
5. Zhang, L.-Q. *et al.* Frequent occurrence of unreduced gametes in *Triticum turgidum*–*Aegilops tauschii* hybrids. *Euphytica* **172**, 285-294 (2010).
6. Allaby, R.G., Stevens, C.J., Kistler, L. & Fuller, D.Q. Emerging evidence of plant domestication as a landscape-level process. *Trends in Ecology & Evolution* (2021).
7. Zhou, Y. *et al.* Triticum population sequencing provides insights into wheat adaptation. *Nature Genetics* **52**, 1412-1422 (2020).
8. Gaut, B.S., Seymour, D.K., Liu, Q. & Zhou, Y. Demography and its effects on genomic variation in crop domestication. *Nature Plants* **4**, 512-520 (2018).
9. Allaby, R. Integrating the processes in the evolutionary system of domestication. *Journal of Experimental Botany* **61**, 935-944 (2010).
10. He, F. *et al.* Exome sequencing highlights the role of wild-relative introgression in shaping the adaptive landscape of the wheat genome (vol 50, pg 896, 2019). *Nature Genetics* **51**, 1194-1194 (2019).
11. Salamini, F., Özkan, H., Brandolini, A., Schäfer-Pregl, R. & Martin, W. Genetics and geography of wild cereal domestication in the near east. *Nature Reviews Genetics* **3**, 429-441 (2002).
12. Matsuoka, Y. & Nasuda, S. Durum wheat as a candidate for the unknown female progenitor of bread wheat: an empirical study with a highly fertile F1 hybrid with *Aegilops tauschii* Coss. *Theoretical and Applied Genetics* **109**, 1710-1717 (2004).
13. Hillman, G. On the origins of domestic rye—*Secale cereale*: the finds from aceramic Can Hasan III in Turkey. *Anatolian Studies* **28**, 157-174 (1978).
14. Araus, J.L., Slafer, G.A., Romagosa, I. & Molist, M. FOCUS: Estimated Wheat Yields During the Emergence of Agriculture Based on the Carbon Isotope Discrimination of Grains: Evidence from a 10th Millennium BP Site on the Euphrates. *Journal of Archaeological Science* **28**, 341-350 (2001).
15. Feldman, M. Origin of cultivated wheat In Bonjean AP, Angus WJ, eds. The world wheat book: a history of wheat breeding. Paris: Intercept. 3-56 (Lavoisier Publishing, 2001).
16. Kihara, H. Morphological, physiological, genetical and cytological studies in *Aegilops* and *Triticum* collected from Pakistan, Afghanistan and Iran. *Results of the Kyoto University Scientific Expedition to the Karakoram and Hindukush, 1955*, 1-118 (1965).
17. Matsuoka, Y. *et al.* Durum wheat cultivation associated with *Aegilops tauschii* in northern Iran. *Genetic Resources and Crop Evolution* **55**, 861-868 (2008).
18. Luo, M.C. *et al.* The structure of wild and domesticated emmer wheat populations, gene flow between them, and the site of emmer domestication. *Theoretical and Applied Genetics* **114**, 947-959 (2007).

19. Oliveira, H.R., Jacocks, L., Czajkowska, B.I., Kennedy, S.L. & Brown, T.A. Multiregional origins of the domesticated tetraploid wheats. *PLoS one* **15**, e0227148-e0227148 (2020).
20. Przewieslik-Allen, A.M. *et al.* The role of gene flow and chromosomal instability in shaping the bread wheat genome. *Nature Plants* **7**, 172-183 (2021).
21. Cheng, H. *et al.* Frequent intra- and inter-species introgression shapes the landscape of genetic variation in bread wheat. *Genome Biology* **20**(2019).
22. Villareal, R.L., Rajaram, S., Mujeeb-Kazi, A. & Del Toro, E. The Effect of Chromosome 1B/1R Translocation on the Yield Potential of Certain Spring Wheats (*Triticum aestivum* L.). *Plant Breeding* **106**, 77-81 (1991).
23. Rosenzweig, B.K., Pease, J.B., Besansky, N.J. & Hahn, M.W. Powerful methods for detecting introgressed regions from population genomic data. *Molecular Ecology* **25**, 2387-2397 (2016).
24. Geneva, A.J., Muirhead, C.A., Kingan, S.B. & Garrigan, D. A New Method to Scan Genomes for Introgression in a Secondary Contact Model. *PLOS ONE* **10**, e0118621 (2015).
25. Martin, S.H., Davey, J.W. & Jiggins, C.D. Evaluating the use of ABBA-BABA statistics to locate introgressed loci. *Molecular biology and evolution* **32**, 244-257 (2015).
26. Price, A.L. *et al.* Sensitive Detection of Chromosomal Segments of Distinct Ancestry in Admixed Populations. *PLOS Genetics* **5**, e1000519 (2009).
27. Racimo, F., Sankararaman, S., Nielsen, R. & Huerta-Sánchez, E. Evidence for archaic adaptive introgression in humans. *Nature Reviews Genetics* **16**, 359-371 (2015).
28. Walkowiak, S. *et al.* Multiple wheat genomes reveal global variation in modern breeding. *Nature* **588**, 277-283 (2020).
29. Brinton, J. *et al.* A haplotype-led approach to increase the precision of wheat breeding. *Communications Biology* **3**, 712 (2020).
30. Todesco, M. *et al.* Massive haplotypes underlie ecotypic differentiation in sunflowers. *Nature* **584**, 602-607 (2020).
31. He, S. *et al.* The genomic basis of geographic differentiation and fiber improvement in cultivated cotton. *Nature Genetics* (2021).
32. Hao, C. *et al.* Resequencing of 145 Landmark Cultivars Reveals Asymmetric Sub-genome Selection and Strong Founder Genotype Effects on Wheat Breeding in China. *Molecular Plant* **13**, 1733-1751 (2020).
33. Wang, Z. *et al.* Genomic footprints of wheat evolution in China reflected by a Wheat660K SNP array. *The Crop Journal* **9**, 29-41 (2021).
34. Haller, B.C. & Messer, P.W. SLiM 3: Forward Genetic Simulations Beyond the Wright-Fisher Model. *Molecular Biology and Evolution* **36**, 632-637 (2019).
35. Haller, B.C., Galloway, J., Kelleher, J., Messer, P.W. & Ralph, P.L. Tree-sequence recording in SLiM opens new horizons for forward-time simulation of whole genomes. *Molecular Ecology Resources* **19**, 552-566 (2019).
36. Kelleher, J., Etheridge, A.M. & McVean, G. Efficient Coalescent Simulation and Genealogical Analysis for Large Sample Sizes. *PLOS Computational Biology* **12**, e1004842 (2016).
37. Dvorak, J., Akhunov, E.D., Akhunov, A.R., Deal, K.R. & Luo, M.-C. Molecular Characterization of a Diagnostic DNA Marker for Domesticated Tetraploid Wheat Provides Evidence for Gene Flow from Wild Tetraploid Wheat to Hexaploid Wheat. *Molecular Biology and Evolution* **23**, 1386-1396 (2006).

38. Murray, M.G. & Thompson, W.F. Rapid isolation of high molecular weight plant DNA. *Nucleic Acids Res* **8**, 4321-5 (1980).
39. Bolger, A.M., Lohse, M. & Usadel, B. Trimmomatic: a flexible trimmer for Illumina sequence data. *Bioinformatics* **30**, 2114-2120 (2014).
40. International Wheat Genome Sequencing Consortium. Shifting the limits in wheat research and breeding using a fully annotated reference genome. *Science* **361**, eaar7191 (2018).
41. Li, H. & Durbin, R. Fast and accurate short read alignment with Burrows–Wheeler transform. *Bioinformatics* **25**, 1754-1760 (2009).
42. Barnett, D.W., Garrison, E.K., Quinlan, A.R., Strömberg, M.P. & Marth, G.T. BamTools: a C++ API and toolkit for analyzing and managing BAM files. *Bioinformatics* **27**, 1691-1692 (2011).
43. Li, H. *et al.* The Sequence Alignment/Map format and SAMtools. *Bioinformatics* **25**, 2078-2079 (2009).
44. McKenna, A. *et al.* The Genome Analysis Toolkit: A MapReduce framework for analyzing next-generation DNA sequencing data. *Genome Research* **20**, 1297-1303 (2010).
45. Cingolani, P. *et al.* A program for annotating and predicting the effects of single nucleotide polymorphisms, SnpEff: SNPs in the genome of *Drosophila melanogaster* strain w1118; iso-2; iso-3. *Fly* **6**, 80-92 (2012).

Reviewers' Comments:

Reviewer #1:

Remarks to the Author:

Nested origin and protracted domestication process of tetraploid wheat uncovered by mosaic ancestry block dissection

There is much that is improved in this second draft, and I appreciate the authors have significantly updated their manuscript in response to reviewers. It is gratifying that the authors agree with so many of the points I raise, including the fundamental one about ancestral diversity explaining the deeper divisions in the intragenomic diversity of the A and B genomes.

The description of the algorithm is improved, I can follow it further, but further explanation is still required. Supplementary Figure 5 now makes more sense, and Figure 1d is improved. However, the reader still cannot follow in Figure 1d how one gets from the pattern observed in the initial binning to the Global AHG assignment. Essential to the description of the algorithm is how the authors are calculating the AHG groups in order to derive AHG diversity (as displayed in Supp Fig 5). The term transition times implies a chronological element, I think the authors mean transition number (ie the number of transitions between AHG's being minimized) if I understand correctly. It is very good to see that the model is verified against simulated data which dramatically increases confidence in it.

Reviewer #2:

Remarks to the Author:

I thank the authors for the detailed answers to my previous comments. My main concerns were addressed: the introduction brings now more information on the state of the art, the text was polished and looks clearer, and figures were simplified.

Reviewer #3:

Remarks to the Author:

This is a much improved version of the paper I read last year. The responses to comments are convincing and I think the authors have succeeded to contribute significantly to the knowledge on the genomics of wheat domestication (although maybe not conclusively). Furthermore, I think the authors gave enough background and perspectives on their results.

The description of IntroBlocker in the text is still hard to follow. On the other hand, Fig. 1d is pretty clear and helped me a lot to understand the proposed method. I think that this method is interesting and potentially a good alternative to others available in the literature. Besides, there is probably room to improve IntroBlocker, for instance, the "hclust" step could be replaced by a k-means procedure since the goal is to make groups. There may also be ways to try different window sizes in Step 1 (which could help to adapt the method to cases with different SNP densities).

The figures are still a problem: they contain too much information. I suggest that each panel (Figs. 1a, 1b, ...) contains only a single plot or multiple plots which are clearly related (e.g., Fig. 1a is OK). For instance, in Fig. 2b the boxplots are not needed (and the text related to in the caption is not clear), and in Fig. 2f the same for the pie chart (which seems to show the same information than the barplot). I understand that the authors have a lot of results and want to show a lot of information, but, to me, these overloaded are a bit erasing their message. In fact, only Fig. 6 is clear!

Finally, at the end of paper, the authors should address these two points: What are the outstanding questions on wheat domestication? What are the challenges ahead for the research on this topic? That

sounds more important than summarizing their results.

There are still some several grammatical errors. I only give a few below.

53: what is "agrio value"?

69: thought -> though

96: delete "the" before "genetic diversity"

I think the authors should explain somewhere around here that A,B, and D are subgenomes.

420: playing -> played

446: keeps -> keep

453: has -> have, "accession" should be plural

Reviewer #1 (Remarks to the Author):

Nested origin and protracted domestication process of tetraploid wheat uncovered by mosaic ancestry block dissection

There is much that is improved in this second draft, and I appreciate the authors have significantly updated their manuscript in response to reviewers. It is gratifying that the authors agree with so many of the points I raise, including the fundamental one about ancestral diversity explaining the deeper divisions in the intragenomic diversity of the A and B genomes.

Response:

We really appreciated the reviewer's valuable comments which greatly help to improve our manuscript.

The description of the algorithm is improved, I can follow it further, but further explanation is still required. Supplementary Figure 5 now makes more sense, and Figure 1d is improved. However, the reader still cannot follow in Figure 1d how one gets from the pattern observed in the initial binning to the Global AHG assignment. Essential to the description of the algorithm is how the authors are calculating the AHG groups in order to derive AHG diversity (as displayed in Supp Fig 5).

Response:

We fully understood the reviewer's concern about the description of the algorithm. To make the global AHG re-assignment process easier to follow, we have integrated its description separated in step 2 and step 3 previously into the new step 2. Accordingly, we have revised figure 1d by incorporating the global AHG re-assignment process as displayed in supplementary figure 5 previously. The difference between the un-supervised mode and semi-supervised mode was shown in the new supplementary figure 5. We revised the description of the algorithm in the main text in line 137 as follows:

Step 1, hierarchical clusters are built based on pairwise genetic distances for each window independently, and initial AHGs are assigned by the hard threshold imputed from the bimodal distribution. Step 2, based on the premise that the ancestry of adjacent genomic regions tends to be the same, AHGs are re-assigned globally to minimize the transition number between adjacent bins under the accession-priority-based AHG referring strategy. For each accession along with the priority order, if its windows are of the same initial AHG type with a high-priority accession, the AHG type of high-priority accession is assigned to the current accession. Otherwise, a novel AHG type is assigned and dominated by the current accession (see Methods

for details). The priority order of accessions is determined by a bubble sort-like method consisting of two parts. Step 2.1 aims to select a subset of accessions covering the majority of AHG diversity (95% in default). Based on this primary order, step 2.2 aims to find the order that could minimize the transition number by swapping accessions iteratively. In each iteration, the order of adjacent accessions is swapped, AHGs are re-assigned using the current order and the transition number is counted. The swapped order is reserved if the transition number decreases, and swapping continues until no changes could be made. This ranking algorithm could be evoked globally (un-supervised mode), and it also could be evoked within each accession group assigned based on prior knowledge (semi-supervised mode, Supplementary Fig. 5). Step 3, a Bayesian method is evoked to smooth the potentially misassigned blocks under the hard threshold by referring to the AHG types in neighboring windows (Supplementary Fig. 6).

Revised Fig 1d, Schematic overview of the IntroBlocker algorithm. Step 1, for each bin, the accessions hierarchically clustered by genetic distance were grouped with a cut-off at the threshold (10^{-3} variants per bp) imputed from a bimodal distribution. Step 2, AHGs were re-assigned globally using an accession priority-based referring strategy to minimize the transition number, where the priority order was determined by a bubble sort-like method. Step 3, a Bayesian approach was introduced for smoothing noisy signals by correcting misassigned blocks based on AHG types of neighboring windows.

Step2: Global AHG re-assignment

New Supplementary Fig. 5. Schema of the un-supervised and the semi-supervised mode of the global AHG re-assignment. In un-supervised mode, the priority order of all accessions is adjusted globally. In semi-supervised mode, the adjustment is confined within each accession group assigned based on prior knowledge.

We revised the detailed description of step 2 of the algorithm in the method section in line 542 as follows:

Step 2, Global AHG re-assignment. AHGs are re-assigned based on the initial AHGs to minimize the transition number between adjacent bins. The re-assignment uses an accession-priority-based AHG referring strategy. For a given priority order, the accession with the highest priority is assigned with a uniform AHG type. For each accession along with this priority order, if its windows are of the same initial AHG type with a high-priority accession, the AHG type of high-priority accession is assigned to the current accession. Otherwise, a novel AHG type is assigned and dominated by the current accession.

The priority order of accessions used in the AHG referring strategy was determined by a bubble sort-like method consisting of two parts. Step 2.1 aims to select a subset of accessions covering the majority of AHG diversity. We implemented an iteratively greedy heuristic algorithm to select one accession in each round. The accession that shares the most initial AHGs with all other accessions is selected in the first round. In the following rounds, the accession that shares the most uncovered AHGs with unselected accessions is selected one by one until most of the initial AHGs could be found in the panel (95% in default). The remaining accessions are randomly appended to the end of the panel.

Step 2.1 aims to find the order that could minimize the transition number by swapping accessions iteratively based on the primary order from step 2.1. In each iteration, adjacent

accessions along the priority order were swapped, AHGs were re-assigned using the current order and the transition number was counted. If the total transition number decreases after swapping, the new order is kept. Otherwise, the original order is maintained. The swapping procedure keeps running until no adjustment could be made.

The re-assignment has two modes (Supplementary Fig. 5). In un-supervised mode, the re-assignment was conducted on all accessions globally. In semi-supervised mode, a pre-defined group could be assigned to each accession based on prior knowledge, and the order of groups should be given, such as the order used in this study “WE>DT>hexaploid wheat”. Within each group, the un-supervised mode is activated internally to generate the priority order independently. The final order is generated by concatenating the order of each group.

The term transition times implies a chronological element, I think the authors mean transition number (ie the number of transitions between AHG's being minimized) if I understand correctly.

Response:

Thanks for the reviewer's comment on the previous term “transition times”. We have changed the term “transition times” to “transition numbers” throughout the manuscript and figure legends.

It is very good to see that the model is verified against simulated data which dramatically increases confidence in it.

Response:

We really appreciated the reviewer's valuable comments which greatly help to improve our manuscript.

Reviewer #2 (Remarks to the Author):

I thank the authors for the detailed answers to my previous comments. My main concerns were addressed: the introduction brings now more information on the state of the art, the text was polished and looks clearer, and figures were simplified.

Response:

We really appreciated the reviewer's valuable comments which greatly help to improve our manuscript.

Reviewer #3 (Remarks to the Author):

This is a much improved version of the paper I read last year. The responses to comments are convincing and I think the authors have succeeded to contribute significantly to the knowledge on the genomics of wheat domestication (although maybe not conclusively). Furthermore, I think the authors gave enough background and perspectives on their results.

Response:

We appreciated the reviewer's acknowledgment of our efforts to improve the manuscript and valuable comments which greatly help to improve our manuscript.

The description of IntroBlocker in the text is still hard to follow. On the other hand, Fig. 1d is pretty clear and helped me a lot to understand the proposed method. I think that this method is interesting and potentially a good alternative to others available in the literature.

Response:

We fully understood the reviewer's concern about the description of the algorithm. To make the global AHG re-assignment process easier to follow, we have integrated its description separated in step 2 and step 3 previously into the new step 2. Accordingly, we have revised figure 1d by incorporating the global AHG re-assignment process as displayed in supplementary figure 5 previously. The difference between the un-supervised mode and semi-supervised mode was shown in the new supplementary figure 5. We revised the description of the algorithm in the main text in line 137 as follows:

Step 1, hierarchical clusters are built based on pairwise genetic distances for each window independently, and initial AHGs are assigned by the hard threshold imputed from the bimodal distribution. Step 2, based on the premise that the ancestry of adjacent genomic regions tends to be the same, AHGs are re-assigned globally to minimize the transition number between adjacent bins under the accession-priority-based AHG referring strategy. For each accession along with the priority order, if its windows are of the same initial AHG type with a high-priority accession, the AHG type of high-priority accession is assigned to the current accession. Otherwise, a novel AHG type is assigned and dominated by the current accession (see Methods for details). The priority order of accessions is determined by a bubble sort-like method consisting of two parts. Step 2.1 aims to select a subset of accessions covering the majority of AHG diversity (95% in default). Based on this primary order, step 2.2 aims to find the order that could minimize the transition number by swapping accessions iteratively. In each iteration,

the order of adjacent accessions is swapped, AHGs are re-assigned using the current order and the transition number is counted. The swapped order is reserved if the transition number decreases, and swapping continues until no changes could be made. This ranking algorithm could be evoked globally (un-supervised mode), and it also could be evoked within each accession group assigned based on prior knowledge (semi-supervised mode, Supplementary Fig. 5). Step 3, a Bayesian method is evoked to smooth the potentially misassigned blocks under the hard threshold by referring to the AHG types in neighboring windows (Supplementary Fig. 6).

Revised Fig 1d, Schematic overview of the IntroBlocker algorithm. Step 1, for each bin, the accessions hierarchically clustered by genetic distance were grouped with a cut-off at the threshold (10^{-3} variants per bp) imputed from a bimodal distribution. Step 2, AHGs were re-assigned globally using an accession priority-based referring strategy to minimize the transition number, where the priority order was determined by a bubble sort-like method. Step 3, a Bayesian approach was introduced for smoothing noisy signals by correcting misassigned blocks based on AHG types of neighboring windows.

Step2: Global AHG re-assignment

New Supplementary Fig. 5. Schema of the un-supervised and the semi-supervised mode of the global AHG re-assignment. In un-supervised mode, the priority order of all accessions is adjusted globally. In semi-supervised mode, the adjustment is confined within each accession group assigned based on prior knowledge.

We revised the detailed description of step 2 of the algorithm in the method section in line 542 as follows:

Step 2, Global AHG re-assignment. AHGs are re-assigned based on the initial AHGs to minimize the transition number between adjacent bins. The re-assignment uses an accession-priority-based AHG referring strategy. For a given priority order, the accession with the highest priority is assigned with a uniform AHG type. For each accession along with this priority order, if its windows are of the same initial AHG type with a high-priority accession, the AHG type of high-priority accession is assigned to the current accession. Otherwise, a novel AHG type is assigned and dominated by the current accession.

The priority order of accessions used in the AHG referring strategy was determined by a bubble sort-like method consisting of two parts. Step 2.1 aims to select a subset of accessions covering the majority of AHG diversity. We implemented an iteratively greedy heuristic algorithm to select one accession in each round. The accession that shares the most initial AHGs with all other accessions is selected in the first round. In the following rounds, the accession that shares the most uncovered AHGs with unselected accessions is selected one by one until most of the initial AHGs could be found in the panel (95% in default). The remaining accessions are randomly appended to the end of the panel.

Step 2.1 aims to find the order that could minimize the transition number by swapping accessions iteratively based on the primary order from step 2.1. In each iteration, adjacent

accessions along the priority order were swapped, AHGs were re-assigned using the current order and the transition number was counted. If the total transition number decreases after swapping, the new order is kept. Otherwise, the original order is maintained. The swapping procedure keeps running until no adjustment could be made.

The re-assignment has two modes (Supplementary Fig. 5). In un-supervised mode, the re-assignment was conducted on all accessions globally. In semi-supervised mode, a pre-defined group could be assigned to each accession based on prior knowledge, and the order of groups should be given, such as the order used in this study “WE>DT>hexaploid wheat”. Within each group, the un-supervised mode is activated internally to generate the priority order independently. The final order is generated by concatenating the order of each group.

Besides, there is probably room to improve IntroBlocker, for instance, the "hclust" step could be replaced by a k-means procedure since the goal is to make groups. There may also be ways to try different window sizes in Step 1 (which could help to adapt the method to cases with different SNP densities).

Response:

Thanks for the reviewer’s valuable suggestions for improving the algorithm. The “hclust” procedure in step 1 of the algorithm is to make groups indeed. But according to the underlying model of our algorithm, it is the genetic distance threshold that defines groups, while the k-means algorithm clusters samples into a pre-defined number of groups. We could not know the exact number of groups of each genetic window, so it is not appropriate to use the k-means algorithm here.

As the reviewer commented, the appropriate window size may vary among species and datasets. Our algorithm supported a user-defined window size. In this study, we set the window size to 5Mb considering the linkage disequilibrium pattern shown in supplementary figure 4. We have revised the method section in line 535 as follows: **For each nonoverlapping genomic window, accessions are clustered with an average-linkage hierarchical clustering algorithm based on the pairwise genetic distance matrix, using the hclust function implemented in R. The size of the window was set to 5 Mb in default, which could be modified according to LD-decay distance in specific species.**

The figures are still a problem: they contain too much information. I suggest that each panel (Figs. 1a, 1b, ...) contains only a single plot or multiple plots which are clearly related (e.g.,

Fig. 1a is OK). For instance, in Fig. 2b the boxplots are not needed (and the text related to in the caption is not clear), and in Fig. 2f the same for the pie chart (which seems to show the same information than the barplot). I understand that the authors have a lot of results and want to show a lot of information, but, to me, these overloaded are a bit erasing their message. In fact, only Fig. 6 is clear!

Response:

We understood the reviewer's concern. Accordingly, we have removed the boxplot from Fig. 2b and the pie chart from Fig. 2f.

Finally, at the end of paper, the authors should address these two points: What are the outstanding questions on wheat domestication? What are the challenges ahead for the research on this topic? That sounds more important than summarizing their results.

Response:

Thanks for the reviewer's valuable suggestion. We have revised the last paragraph of the discussion section in line 466 as follows:

As climate change accelerated, harnessing the adaptation mechanism of major crops to create outstanding cultivars has become necessary and urgent⁶⁶. A greater understanding of domestication would provide a theoretical basis for how we could achieve it⁶⁷, like creating novel crops through *de novo* domestication, especially polyploids of vigorosity and robustness⁶⁸. Besides the knowledge regarding wheat domestication gained during the last decade, still many questions remain unsettled. For example, the formation mechanisms and genetic relationships of subspecies in the *Triticum-Aegilops* complex could be illuminated further in the light of ancestral mosaics. Considering the current report about the reticulated evolution and frequent interploidy introgression of the bread wheat³, the synergistic improvement of both tetraploid durum wheat and hexaploid bread wheat could be accelerated by transferring the beneficial alleles between the two genetic pools. Furthermore, a large portion of domesticated genes within the favorable genomic segments during domestication and improvement was underexplored, and their detailed evolutionary trajectories remain unclear. Additionally, the relationship between the gene pool of *Aegilops tauschii* and the wheat D subgenome may be oversimplified⁶². With comprehensive genomic data of *Triticum-Aegilops* species available and innovations in computing algorithms, answers to these questions will be clear and ultimately used as a source of innovations for wheat improvement.

There are still some several grammatical errors. I only give a few below.

53: what is "agrio value"?

Response:

We are sorry for this apparent mistake. We have corrected this sentence as “some of which were valuable for agronomic practices.” in line 53.

69: thought -> though

Response:

We have revised the sentence in line 69.

96: delete "the" before "genetic diversity"

Response:

We have revised the sentence in line 95.

I think the authors should explain somewhere around here that A,B, and D are subgenomes.

Response:

Thanks for the reviewer’s valuable suggestion. We have added a sentence in line 94 as follows: Among the A, B, and D subgenomes of wheat, introgressions have been reported to be pervasive in the A&B subgenome, and it is positively correlated with genetic diversity.

420: playing -> played

Response:

We have revised the sentence in line 424.

446: keeps -> keep

Response:

We have revised the sentence in line 450.

453: has -> have, "accession" should be plural

Response:

We have revised the sentence in line 457.

Reviewers' Comments:

Reviewer #1:

Remarks to the Author:

In this third draft of the manuscript the authors really have done an excellent job of making the algorithm methodology clearer. I can now see their method and confidently get behind it. I commend the authors on their willingness to meet reviewer demands. I think this is a valuable contribution and now ready for publication.

Reviewer #3:

Remarks to the Author:

I congratulate the authors for their work.

Reviewer #1 (Remarks to the Author):

In this third draft of the manuscript the authors really have done an excellent job of making the algorithm methodology clearer. I can now see their method and confidently get behind it. I commend the authors on their willingness to meet reviewer demands. I think this is a valuable contribution and now ready for publication.

Response:

We really appreciated the reviewer's valuable comments which greatly help to improve our manuscript.

Reviewer #3 (Remarks to the Author):

I congratulate the authors for their work.

Response:

We really appreciated the reviewer's valuable comments which greatly help to improve our manuscript.